# PTMNavigator: interactive visualization of differentially regulated post-translational modifications in cellular signaling pathways

Julian Müller [1], Florian P. Bayer[1], Mathias Wilhelm [2], Maximilian G. Schuh [1,4], Bernhard Kuster [1,3] & Matthew The [1] ✉

Post-translational modifications (PTMs) play pivotal roles in regulating cellular signaling, fine-tuning protein function, and orchestrating complex biological processes. Despite their importance, the lack of comprehensive tools for studying PTMs from a pathway-centric perspective has limited our ability to understand how PTMs modulate cellular pathways on a molecular level. Here, we present PTMNavigator, a tool integrated into the ProteomicsDB platform that offers an interactive interface for researchers to overlay experimental PTM data with pathway diagrams. PTMNavigator provides ~3000 canonical pathways from manually curated databases, enabling users to modify and create custom diagrams tailored to their data. Additionally, PTMNavigator automatically runs kinase and pathway enrichment algorithms whose results are directly integrated into the visualization. This offers a comprehensive view of the intricate relationship between PTMs and signaling pathways. We demonstrate the utility of PTMNavigator by applying it to two phosphoproteomics datasets, showing how it can enhance pathway enrichment analysis, visualize how drug treatments result in a discernable flow of PTM-driven signaling, and aid in proposing extensions to existing pathways. By enhancing our understanding of cellular signaling dynamics and facilitating the discovery of PTM-pathway interactions, PTMNavigator advances our knowledge of PTM biology and its implications in health and disease.

Protein post-translational modifications (PTMs) are among the key regulators of cellular pathways. The presence of a certain PTM on a specific residue can, for instance, determine whether or not the protein can interact with other proteins, activate or inactivate an enzyme, alter a protein's location in the cell, or mark it for degradation. Understanding the specific role of various types of PTMs within intracellular signaling cascades is essential to gain insights into the molecular mechanisms that govern biological processes. Perturbation studies that examine the effects of altered PTM patterns on pathways have been particularly useful in advancing our knowledge[1–7]. While proteome-wide measurements of PTMs have become a routine procedure for proteomics laboratories, the correct interpretation of the data in the context of cellular signaling pathways continues to pose a challenge.

Multiple databases dedicated to the study of pathways are available to the scientific community[8–11], including the Kyoto Encyclopedia of Genes and Genomes (KEGG) and WikiPathways. KEGG[9] contains 563 manually curated pathway maps with annotations for almost 9000 organisms. WikiPathways[10] is an open-source initiative to collect, maintain, and disseminate data on biological pathways. Its March 2023

[1]Proteomics and Bioanalytics, School of Life Sciences, Technical University of Munich, Freising, Germany. [2]Computational Mass Spectrometry, School of Life Sciences, Technical University of Munich, Freising, Germany. [3]German Cancer Consortium (DKTK), Partner Site Munich, Munich, Germany. [4]Present address: Organic Chemistry II, School of Natural Sciences, Technical University of Munich, Garching, Germany. ✉e-mail: matthew.the@tum.de

release consists of 3130 pathways across 33 species. Both databases are designed to study relationships between genes, gene products, and compounds (such as metabolites or drugs). They offer no (KEGG) or only limited (WikiPathways) direct support for research on the role of specific PTM sites within pathways. For the study of PTM data, several resources exist, the most notable being PhosphoSitePlus[12], which includes data from a large number of research articles not only on phosphorylation but also on acetylation, ubiquitinylation, and other PTMs. Where evidence exists, the sites in PhosphoSitePlus are annotated with their functions and putative upstream or downstream interactors or modifiers. While PhosphoSitePlus has been a tremendously helpful resource, it remains tedious to explore multiple or all sites of a signaling cascade together in order to uncover potential relationships between them and perform higher-level analysis of the data.

A commonly employed method to place all regulated peptides into the context of signaling networks is pathway enrichment analysis (PEA)[13–15], implemented in tools such as g:Profiler[16]. While very useful for the analysis of proteome expression changes, this approach has several shortcomings in the context of PTM analysis, as has been pointed out by Krug et al.[17]. The authors introduced two algorithms that address the problems of traditional PEA: One is gene-centric redundant (GCR) single-sample gene set enrichment analysis (ssGSEA), in which genes are potentially counted multiple times during the calculation of enrichment scores, depending on how many regulated PTMs they harbor. The second is PTM-signature enrichment analysis (PTM-SEA), which uses a specifically assembled database of associations between sites of PTMs and perturbations, pathways, or kinase activities (PTM signatures database; PTMsigDB) and thus can calculate enrichments on PTM level. This constitutes a valuable complementary approach to conventional PEA; both methods have shown superior performance to non-redundant gene-centric PEA. A number of software tools to visualize PTM datasets and pathways have been developed[18–23]. For instance, Phosphomatics[23] is a web service that helps researchers examine kinase-substrate relationships in their phosphoproteomic datasets, whereas KEGGViewer[19] can integrate gene expression profiles and KEGG pathways. A functionality that these tools have been lacking so far is the visualization of regulated PTMs directly within biological networks. The Cytoscape[18] app PhosphoPath[21] has addressed this issue for time-course PTM data, but the app is no longer supported by Cytoscape. PHONEMeS[22] is a software that creates data-driven pathways from phosphoproteomics datasets. Starting from a prior knowledge network of protein-protein and kinase-substrate interactions, it uses mathematical optimization to extract the subset of interactions that best explain the observations. However, running the software and visualizing the results is not trivial for non-expert users.

To overcome the aforementioned problems of pathway enrichment on the PTM level and to empower researchers to visually examine pathway engagement via PTM regulation, we introduce PTMNavigator, a web application that projects PTM perturbation datasets onto pathway diagrams in the form of interactive graphs. To this end, we created internal representations in ProteomicsDB for thousands of pathways from public databases. This enables users to trace signaling cascades across a cell and identify pivotal PTMs within a pathway. PTMNavigator is available as part of the 'Analytics Toolbox' of ProteomicsDB[24,25] and can visualize data hosted on ProteomicsDB as well as user-uploaded data. In addition, we implemented the functionality to design custom pathway diagrams within PTMNavigator in order to create bespoke representations of any PTM dataset. The core visualization component of our software can be easily reused in web interfaces outside of ProteomicsDB. Moreover, we implemented a service that provides programmatic access to several PTM analysis algorithms, including GCR-ssGSEA, PTM-SEA, and PHONEMeS[17,22,26–31]. This service, which can also be used independently, is automatically

invoked when a dataset is uploaded, and the results are integrated into PTMNavigator's user interface. We demonstrate the utility of our application using two phosphoproteomic drug profiling studies[2,32]. First, we show how PTMNavigator improves the interpretation of PEA by integrating the results of multiple enrichment algorithms with visualizations of all potentially relevant pathway diagrams. Second, we visualize target and pathway engagement of kinase inhibitors at different steps in a phosphorylation-dependent signaling cascade. Last, we show how PTMNavigator leverages the information contained in dose-dependent drug perturbation studies to aid the functionalization of PTM sites.

## Results

### A PTM-centric interface for pathway-level data analysis

PTMNavigator is a web-based graphical interface to analyze the results of PTM perturbation experiments in the context of pathways (Fig. 1A). It combines interactive visualization and various enrichment analysis algorithms (Fig. 1B) into a single application. The interface is embedded into the ecosystem of ProteomicsDB, a multi-omics resource for life science data and data analysis. The software combines two types of inputs. The first input, which needs to be provided by the user, is a list of regulated modified peptides, e.g., the result of a differential or dose-response analysis. Each peptide must be annotated with a regulation type (i.e., not, up, or down). The second type of input is pathway diagrams. These can be of two different types: first, users can display canonical pathways from KEGG and WikiPathways, for which we created internal representations in ProteomicsDB. For *Homo sapiens*, we imported 1121 pathways (779 from WikiPathways and 342 from KEGG) from the latest releases at the time of retrieval (see Methods). In addition, we imported 1838 pathways for 9 model organisms, including other mammals, bacteria, and plants (Supplementary Fig. 1). Second, users can also create their own diagrams by either modifying the canonical pathways or by drawing custom diagrams from scratch using the "Editing Mode" of PTMNavigator. The custom diagrams can be stored in ProteomicsDB and shared with other researchers.

PTMNavigator combines the experimental data and the pathway information into a projection of modified peptides onto their corresponding genes within a pathway diagram. The main interface of PTMNavigator (Fig. 2A) is accessible at www.proteomicsdb.org/analytics/ptmNavigator. Users can visualize datasets already contained in ProteomicsDB or upload their own (templates for input formats are available in Supplementary Data 1). It is possible to access (previously) uploaded datasets by a private, user-specific session ID and to visualize one or multiple datasets at a time. When the user selects a pathway diagram, it is rendered with additional nodes for each PTM peptide next to its corresponding protein. These additional PTM nodes are colored according to their regulation (up, down, not). The initial coordinates of the nodes are the same as those in the reference databases, so they resemble the original pathway diagram. However, users can rearrange the layout by dragging nodes around, highlighting interesting subnetworks, and filtering their data by experiments or regulation categories. Where available, additional information on PTM sites is displayed as part of a tooltip, and kinase-substrate relationships can be visualized in the form of additional arrows in the diagram. To this end, we imported a list of functional annotations and upstream regulators from PhosphoSitePlus and a list of functional scores predicted by Ochoa et al.[33] into ProteomicsDB. It is also possible to visualize protein-level data in combination with PTM-level data (see Supplementary Fig. 5 for an example, where this has been exemplified using a dataset from Lee et al.[34]).

In addition to the pathway visualization component, we implemented a service that hosts a number of established enrichment algorithms (Fig. 1B). After a dataset is uploaded, this service is automatically invoked, and two pathway enrichment analysis algorithms

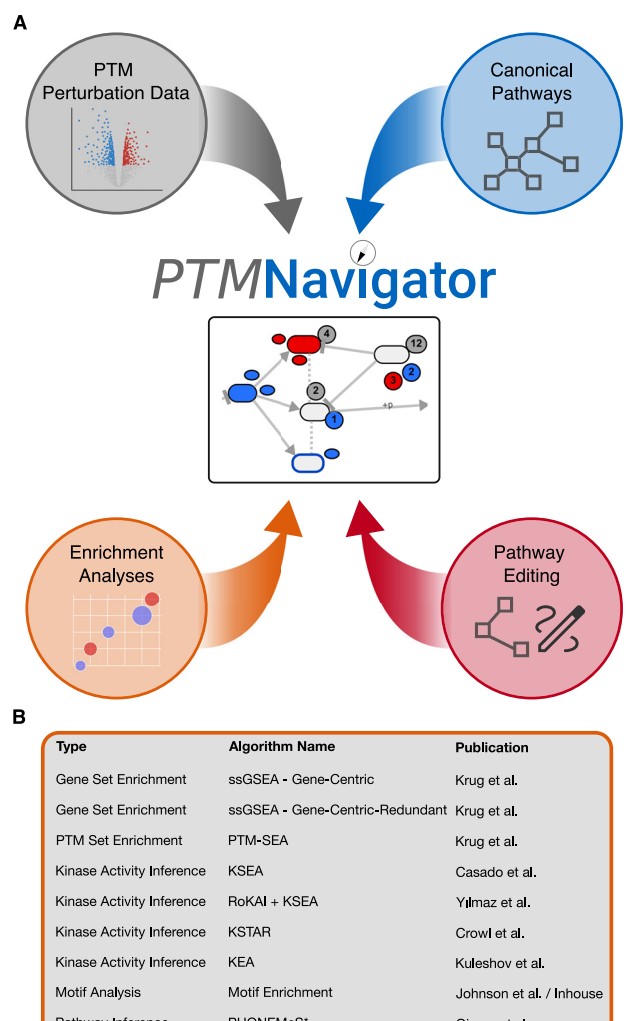

**A**

PTM Perturbation Data

Canonical Pathways

*PTM*Navigator

Enrichment Analyses

Pathway Editing

**B**

| Type | Algorithm Name | Publication |
|---|---|---|
| Gene Set Enrichment | ssGSEA - Gene-Centric | Krug et al. |
| Gene Set Enrichment | ssGSEA - Gene-Centric-Redundant | Krug et al. |
| PTM Set Enrichment | PTM-SEA | Krug et al. |
| Kinase Activity Inference | KSEA | Casado et al. |
| Kinase Activity Inference | RoKAI + KSEA | Yılmaz et al. |
| Kinase Activity Inference | KSTAR | Crowl et al. |
| Kinase Activity Inference | KEA | Kuleshov et al. |
| Motif Analysis | Motif Enrichment | Johnson et al. / Inhouse |
| Pathway Inference | PHONEMeS* | Gjerga et al. |

**Fig. 1 | PTMNavigator combines PTM perturbation data, pathway diagrams, and enrichment analyses within a single application. A** The four building blocks of PTMNavigator. Data from PTM perturbation experiments is provided by the user (top left). This data can then be projected onto canonical pathway diagrams from KEGG and WikiPathways (top right). Uploaded datasets are automatically processed using several enrichment analysis algorithms and the results are integrated into the visualization and displayed in tabular form (bottom left). The user also has the option to design custom pathways, either by using the canonical diagrams as templates or by starting from scratch (bottom right). Parts of the figure were created in BioRender[53]. **B** Overview of the enrichment algorithms available in PTMNavigator. *PHONEMeS is not an enrichment algorithm. Instead, it is a method to reconstruct pathways for phosphoproteomics datasets using prior knowledge.

are run over the dataset ("Gene-Centric" and "Gene-Centric-Redundant" ssGSEA). For phosphoproteomics datasets, additional algorithms are run, including PTM-SEA, KSEA, and PHONEMeS. The results of the enrichments are displayed in tabular form in PTMNavigator (Fig. 2B). In the pathway selection menu in PTMNavigator, pathways are sorted by "Gene-Centric-Redundant" ssGSEA scores to show to the users which pathways are likely most relevant for the dataset. Some of the information is also integrated into the pathway view, e.g., for phosphoproteomics datasets, inferred kinase activities can be shown as colored borders around the nodes in the graph.

The final pathway diagrams, as well as the results of the enrichment analysis, can be downloaded as scalable vector graphics (SVG) or in tabular format, respectively. In the following, we exemplify the capabilities and benefits of PTMNavigator using two recent PTM perturbation datasets.

## Enrichment analysis and pathway visualization synergize in PTMNavigator

To demonstrate how PTMNavigator aids in interpreting the results of a pathway enrichment analysis (PEA), we re-analyzed data from a recent phosphoproteomics study, in which 30 different kinase inhibitors targeting members of the EGF signaling pathway (Fig. 3A) were profiled for their effect on the phosphoproteome of a human retinal pigment epithelial cell line (RPE1)[32]. Re-analysis of all 30 datasets yielded 4989 regulations (2563 up and 2426 down) on 1757 peptides (out of 7871 measured in each sample) at 10% FDR. PEA for each compound with gene-centric-redundant ssGSEA resulted in enrichment scores (negative log-transformed *p*-values; ES) for each combination of drug and pathway (Fig. 3B). Some results confirmed prior knowledge on the compounds, e.g., the six drugs inhibiting either EGFR or MEK, which are both located upstream of MAPK, were strongly enriched for "WP: MAPK Cascade". A PTM-SEA analysis of the same dataset (Supplementary Fig. 6) showed the datasets of the mTOR inhibitors Everolimus and Temsirolimus to be enriched for both the signatures "PERT-PSP_Rapamycin" and "PERT-P100-DIA2_Sirolimus". Rapamycin and Sirolimus are different names for the same compound that also inhibits the kinase mTOR. Yet, for many drug perturbations, the results were not as easy to interpret. For example, the KEGG pathway "Progesterone-Mediated Oocyte Maturation" received a significant score in three of the datasets; however, since the cells studied were retinal cells, it is not apparent where this significance arose from.

To gain more insight into how the regulated peptides were distributed within the enriched pathways, we used PTMNavigator to investigate the high-scoring pathways of the pan-PKC inhibitor Go-6983, one of the datasets which was enriched in "Progesterone-Mediated Oocyte Maturation". Other pathways that received high Enrichment Scores in the ssGSEA analysis of this dataset (Fig. 3C) include signaling networks related to prostate cancer and the nervous system. Manual inspection of the pathways showed that genes with strong fold changes are often shared. Most of the high-scoring pathways contain 4E-BP1, RPS6KA, MAPK1, MAP3K7, mTOR, and/or AKT, all of which have peptides with strong fold changes in the Go-6983 data (Supplementary Fig. 7). Therefore, these proteins likely were the drivers of the high enrichment scores. In the PTM-SEA analysis of the dataset (Fig. 3D), the highest positively and negatively enriched signatures, respectively, were the mTOR and RPS6KA1 PhosphoSitePlus signatures (Scores = 5.78 and −5.76, FDR = 0.022 for both). mTOR and RPS6KA1 were also reported to have increased (decreased) activity by KSEA. We investigated these scores in more depth, using the PhosphoSitePlus kinase-substrate annotations integrated into PTMNavigator (Supplementary Fig. 8). Highlighting the substrates of mTOR and RPS6KA1 within the diagram "mTOR Signaling Pathway - hsa04150" showed that most of the peptides phosphorylated by RPS6KA1 had substantial negative fold changes (e.g. EIF4B_S422 and SOS1_S1134). In contrast, most mTOR substrates were increased in phosphorylation (e.g., the autophosphorylation sites T2474 and S2478). This demonstrates how PTMNavigator can help interpret the results of enrichment analyses.

## Visualizing target and pathway engagement in PTMNavigator

Next, we employed PTMNavigator on the same dataset to compare the kinase inhibitors Lapatinib, Wortmannin, and Temsirolimus, which target the same signaling cascade but at different steps. Projection of their perturbed PTM peptides on "mTOR signaling pathway – hsa04150" (Fig. 4A), which includes all members of said cascade except for EGFR, confirmed that all three drugs perturbed the pathway. Lapatinib had the largest number of up- and downregulated peptides and the most upstream regulations in the pathway (Fig. 4A, upper panel). Among the sites exclusively regulated by Lapatinib were GSK3B_S9, TSC2_S939, and BRAF_S729, whose importance has previously been pointed out in the literature[35–37]. In contrast, the effects of Wortmannin in this pathway were limited to sites on PRAS40, LPIN, 4E-

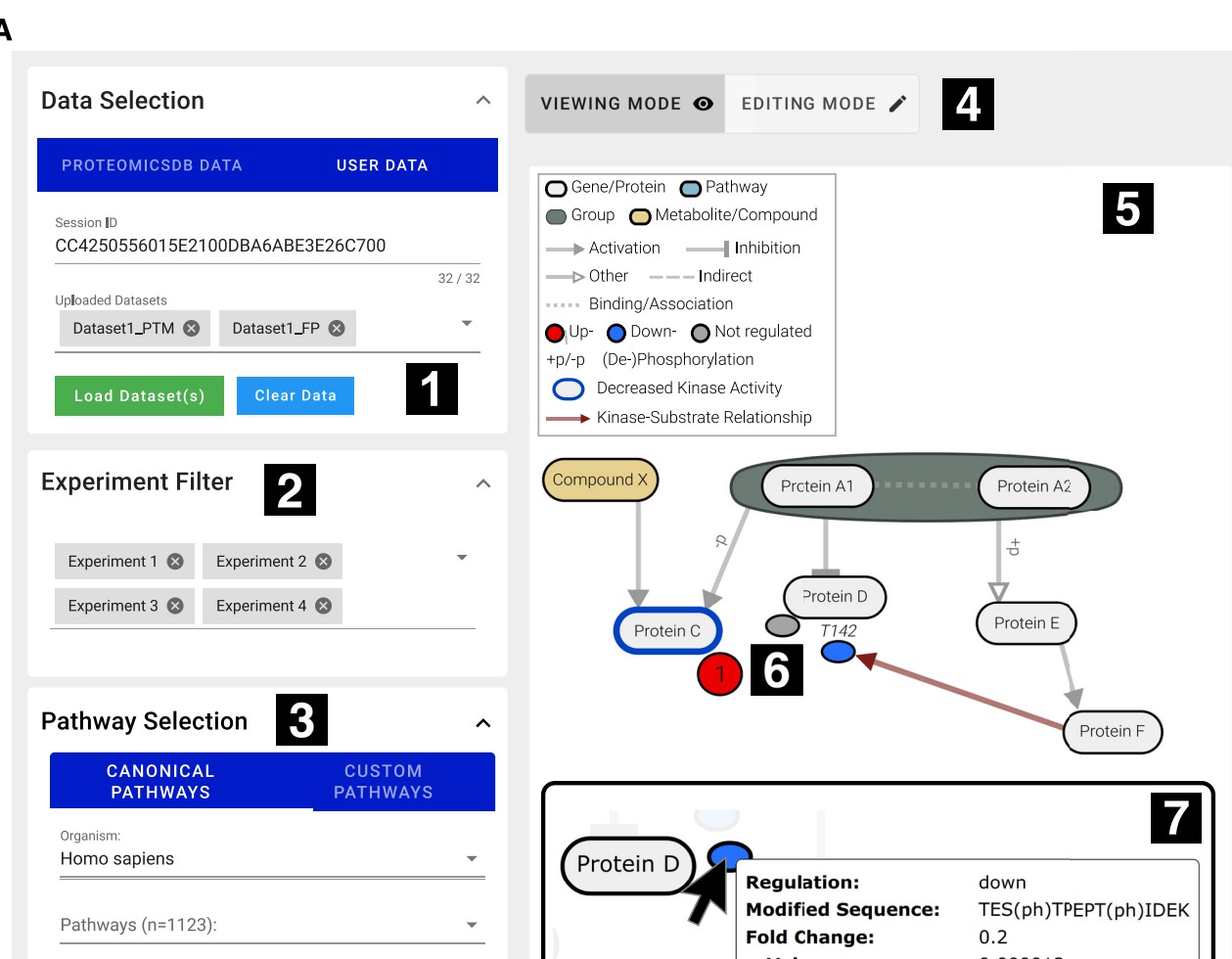

BP1, and RPS6, all of which are downstream of PI3K in the depicted network (Fig. 4A, middle panel). Likewise, Temsirolimus regulated even fewer sites, and all are downstream of mTOR (Fig. 4A, lower panel). Most of the regulated peptides for Wortmannin and Temsirolimus were shared with Lapatinib (PRAS40_T246, 4EBP1_T77/S101, LPIN1_S252 for Wortmannin, RICTOR_Y1174&Y1177 for Temsirolimus), or one another (RPS6_S236&S240&S244). This was further substantiated by the results of kinase activity scoring on the three datasets: KSEA produced significant negative scores for AKT, RAF, RPS6KA, and RPS6KB in the Lapatinib and Wortmannin data. For Temsirolimus, only RPS6KA and RPS6KB were reported to have decreased activity. RPS6KB is the only of these kinases that appear downstream of mTOR, and the high score of RPS6KA probably stems from the large number of substrates shared with RPS6KB. These observations confirm that all drugs perturb the same signaling network at different stages.

**Fig. 2 | The user interface of PTMNavigator. A** The main interface of PTMNavigator. (1) Datasets can be accessed via a 32-digit Session ID that users receive during data upload. An arbitrary combination of datasets can be loaded at the same time. (2) A dataset may contain multiple experiments (specified in the uploaded CSV file). In this panel, experiments can dynamically be added to and removed from the diagram. (3) The list of pathways to choose from is sorted by scores from a gene-centric redundant enrichment analysis. Users can also search this list for the name or identifier of a pathway or filter for pathways that contain a specific set of genes or proteins. (4) Users can switch between a viewing mode, in which they can study the projection of their data onto pathway diagrams, and an editing mode, in which they can modify existing and create custom pathways. (5) The main canvas. After

selecting a pathway, its diagram is rendered here. Additional smaller nodes are added for the peptides in the user data. (6) Initially, all peptides of a regulation group (up-/down-/unregulated) belonging to the same protein are summarized in one circular node labeled with the number of peptides it contains. Double-clicking on it allows a summary node to be expanded into individual nodes for each peptide (elliptical shapes). The position of the modified residue can optionally be displayed above the ellipsis. (7) All information supplied during the upload is displayed as a tooltip when hovering over a node. If a site is annotated in PhosphoSitePlus, a link to the annotation is retrieved. **B** The Enrichment Analyses View. Here, it shows the three most significant results from a Kinase-Substrate Enrichment Analysis (KSEA).

A more surprising example was presented when comparing the effects of Sorafenib and Cobimetinib. Both compounds target the RAF-MEK-ERK pathway, another phosphorylation axis downstream of EGFR (Fig. 3A). Since Sorafenib is perturbing the network at a more upstream position (RAF), one would expect its impact on the PTM level to be more pronounced. In reality, Cobimetinib (which inhibits MEK) had a broader effect than Sorafenib, which was visible in PTMNavigator's visualization of "MAPK Signaling Pathway - hsa04010" (Fig. 4B). As expected, Cobimetinib significantly inhibited the phosphorylation of the site MAPK1_Y187, which is a MEK substrate and essential to MAPK1 activity, with a Log2 Fold Change (logFC) of −4.39 between conditions (p-value = 0.01). There also were effects upstream in the pathway, e.g., on SOS1_S1134. This is likely the result of an inhibited feedback loop since this site is a substrate of RPS6KA1, which is regulated by ERK/MAPK[38]. In contrast, Sorafenib only inhibited the phosphorylation of one site in this pathway (BRAF_S729), which was also strongly downregulated by Cobimetinib (logFC = −5.44, p = 0.07 for Sorafenib and logFC = −5.41, p = 0.09 for Cobimetinib) and is known to be essential for BRAF-dimerization[39]. PTMNavigator's visualization suggests that Sorafenib does engage its target, but the perturbation in signaling does not progress beyond RAF. Previous research on the compound pointed out that Sorafenib, while an effective agent against renal cell carcinoma and other cancer types, might not exert its efficacy via inhibition of RAF[40,41]. In both examples presented here, PTMNavigator provided a visual insight into the target engagement and signaling cascades that one would not get as quickly from other depictions such as heatmaps.

### Exploring dose-dependent perturbation data in PTMNavigator

A fundamental characteristic of a drug's mode of action is how potently it perturbs different parts of cellular signaling, which can be captured by applying a series of doses of the compound to a model system and measuring the PTM-level changes at each dose. Such a concentration-dependent approach, called decryptM, was recently published by Zecha et al.[2]. decryptM allows the determination of dose-response characteristics for each peptide, including drug potency (negative logEC50 or pEC50 value) and drug effect size (curve fold change). Output files of the decryptM pipeline[2] or CurveCurator[42] can directly be imported by PTMNavigator. We implemented additional features for dose-dependent data, which we demonstrate by using the tool to compare data from Zecha et al. to data from Bekker-Jensen et al. for the EGFR inhibitor Lapatinib.

The decryptM study included an experiment in which Lapatinib was applied to the breast cancer cell line MDA-MB-175-VII for 2 h in ten different concentrations (from 1 nM to 10 μM). The experiment yielded 180 upgoing and 441 downgoing dose-response curves classified as significant (see Methods). PTMNavigator's projection of the regulated peptides onto the pathway "mTOR signaling pathway - hsa04150" (Fig. 5A; this was the highest-scoring pathway in the gene-centric and gene-centric-redundant ssGSEA of the dataset) showed that the overall perturbation of the network was very similar to that observed in the Lapatinib experiment by Bekker-Jensen et al. on RPE1 cells (Fig. 4A, upper panel). Many proteins had regulated peptides in both

experiments (for example, RAF, MEK, and RICTOR), and some downregulated sites were shared by both biological systems (TSC2_S939, PRAS40_T246, and GSK3B_S9). However, the decryptM data adds the potency dimension for every regulation as an additional layer of information (Fig. 5B), represented in PTMNavigator by PTMs colored according to their pEC50 value. This visualization reveals that all peptides perturbed by Lapatinib in this pathway were regulated at a pEC50 between 6.4 and 7.4, which corresponds to a concentration range between 40 nM and 400 nM (Fig. 5C). The dose-response curves of each shown peptide can be displayed directly within PTMNavigator. Lapatinib is known to be a highly selective EGFR/ErbB2 inhibitor[40], and the fact that all potencies were within one order of magnitude suggests that this pathway is only regulated by a single initial perturbation. Here, this is likely the inhibition of ErbB2, as EGFR has been reported to be inactive in MDA-MB-175-VII cells[43]. In other biological systems, the effect might differ. Such cell-type specific observations can also be highlighted by PTMNavigator, which we exemplified by comparing the impact of Dasatinib on three different cell lines (Supplementary Fig. 9).

### PTMNavigator aids in the refinement and expansion of pathway diagrams

We applied PTMNavigator to two other decryptM datasets to put phosphorylation sites with and without functional annotation into context. Refametinib and Mirdametinib are two highly selective kinase inhibitors that both target MAP2K1 (MEK1) and MAP2K2 (MEK2). Both inhibitors were applied to A549 lung carcinoma epithelial cells in the decryptM study (in ten concentrations ranging from 1 nM to 10 μM). Both drugs elicited a selective response in the A549 phosphoproteome, with 148 regulated dose-response curves for Refametinib (38 up, 110 down) and 73 for Mirdametinib (15 up, 58 down). Among the enriched pathways for both datasets, the diagram "Regulation of actin cytoskeleton – WP51" depicted the signaling around the compounds" targets most comprehensively. Projection of the drug-regulated peptides onto this pathway using the "potency" color scheme (Fig. 6A) showed that Mirdametinib regulates MEK signaling more potently than Refametinib. This is in agreement with the reported higher affinity observed in pulldown experiments[40].

By focusing on the common dose-dependent phosphorylation signals between the two drugs, we obtained a more complete understanding of the MEK signaling in this cellular system. Both drugs potently downregulated phosphorylation of MAPK1 and MAPK3 at their most important activating sites (T185 & Y187 for MAPK1, T202 & Y204 for MAPK3), which are all direct substrates of MEK1 and MEK2 and are part of the MEK1 kinase activity signature in PTMSigDB[17]. Downstream of MAPK, both drugs elicited downregulation of several peptides, including MYPT1_S507, a site that has previously been associated with treatment by Selumetinib (a MEK1 and MEK2 inhibitor)[44] and Vemurafenib (which has MEK5 as a potent off-target)[40,45]. The fact that it is also affected by the two inhibitors studied here and that its dose-response profiles are similar to other peptides associated with MEK inhibition (Supplementary Fig. 10) suggests that this site could also be part of the general PTM signature of MEK inhibition. Less straightforward to interpret were several

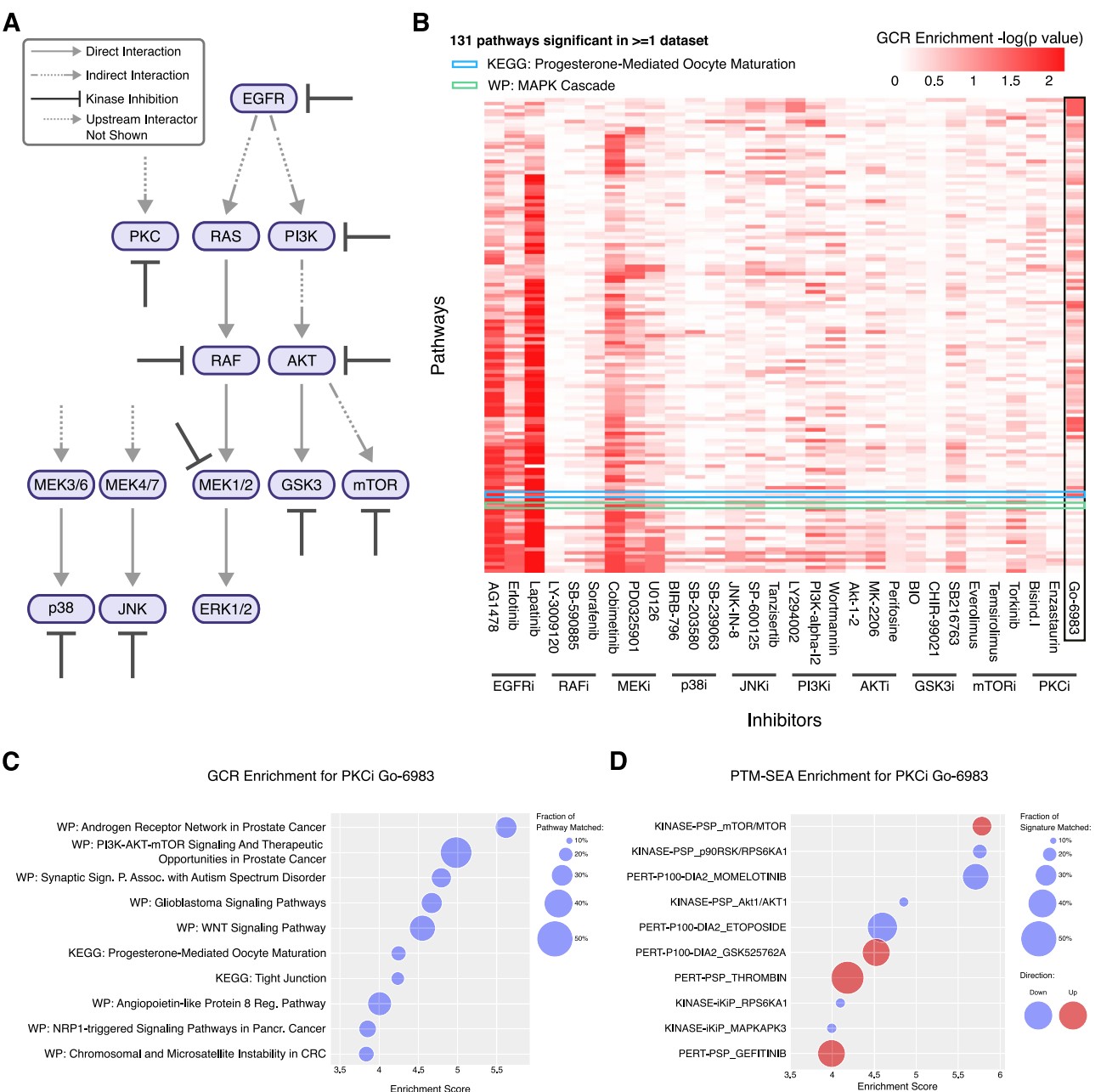

**Fig. 3 | PTMNavigator aids in interpreting pathway enrichment analysis results.** **A** Overview of the kinases that were targeted with inhibitors in the study by Bekker-Jensen et al.[32] **B** Results of gene-centric-redundant ssGSEA showing pathways with *p*-value < 0.05 for the 30 kinase inhibitor datasets. Pathways are sorted by the number of datasets in which they were reported as enriched (larger numbers on the bottom). Inhibitors are grouped by their designated targets. *p*-values were corrected for multiple testing using the Benjamini-Hochberg procedure. The plot was generated using Python. GCR: gene-centric redundant. **C** 10 highest-scoring pathways for the PKC inhibitor Go-6983 by gene-centric-redundant ssGSEA score. Circle size indicates the fraction of genes in the pathway contained in the set of regulations. The plot was generated using Python. **D** 10 highest-scoring pathways for the PKC inhibitor Go-6983 by absolute PTM-SEA score. Circle size indicates the fraction of genes in the signature that were contained in the set of regulations; color indicates the direction of regulation. The plot was generated using Python. Source data for (B, C, D) are provided as a Source Data file.

downregulations upstream of MEK at SOS1 sites as well as RAF1_S43 and GRLF1_S589 in both datasets. This is an example where the pathway diagram does not depict all relevant interactions for the cellular system, a frequently observed issue of the canonical pathway definitions (see Discussion). To obtain a more complete network of the effects of MEK inhibition, we investigated the pathway that PHONEMeS generated for the dataset (Fig. 6B) and consulted previous research on the matter. PHONEMeS reproduced the RAF-MAP2K-MAPK axis that is the essential part of the network. It also proposed a flow of signal from MAPK to SOS via RPS6KA and from

MAPK to RAF1. Prior literature confirmed the first proposal: the regulated SOS1 residues S1134 and S1161 are RPS6KA substrates[38], and RPS6KA activity is controlled by MAPK[46]. Therefore, this regulation likely is the result of an inhibited feedback loop. SOS1_S1064 has no annotated kinase, but the other regulations suggest it could also be an RPS6KA substrate. RAF1_S43, however, is a substrate of PKACA, not MAPK[47]. The "Editing Mode" of PTMNavigator allowed us to integrate this information into an extended pathway, using the diagram "Regulation of actin cytoskeleton – WP51" as a starting point (Fig. 6C). GRLF_S589 has no functional annotation or putative

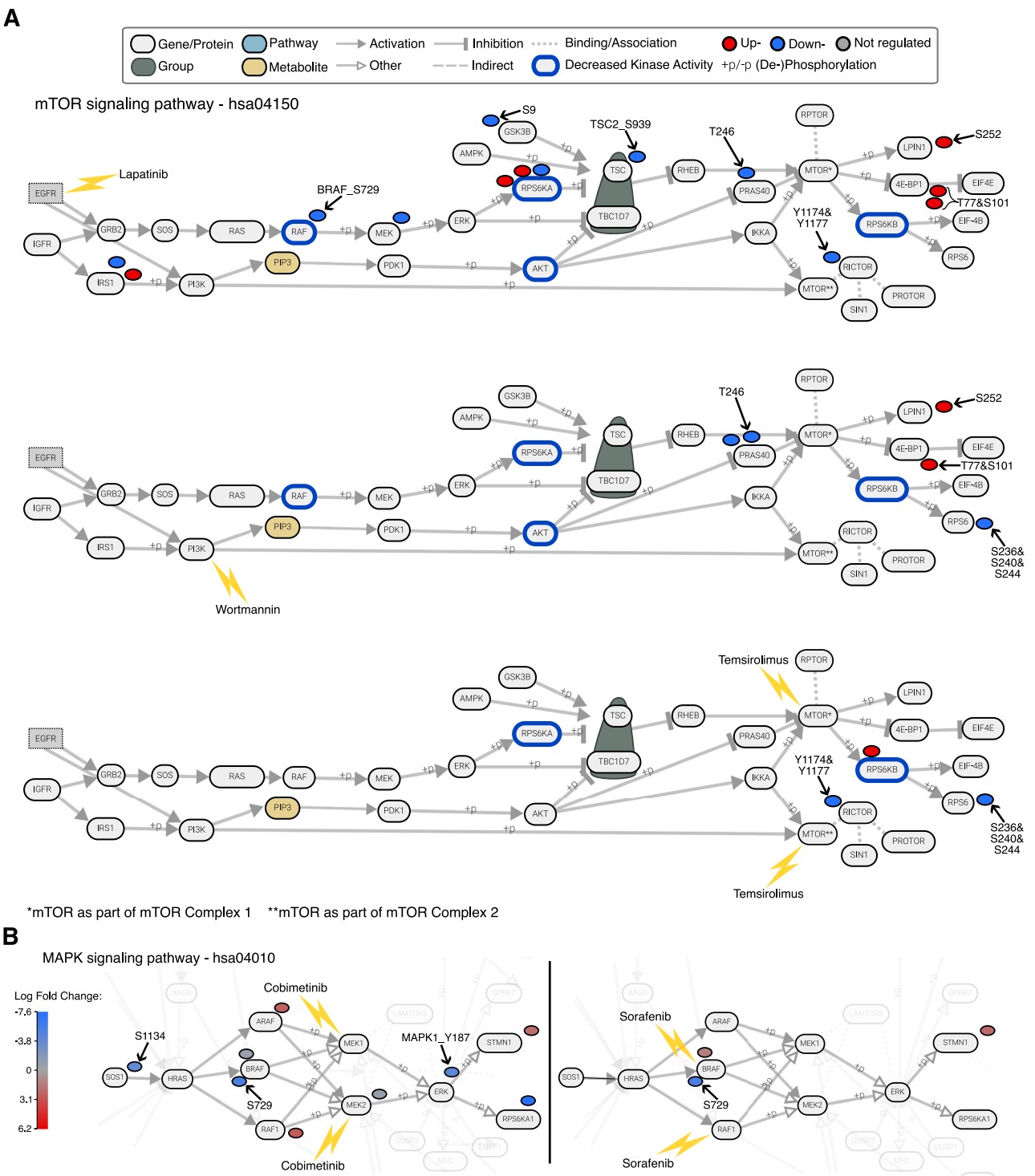

**Fig. 4 | Analysis of a phosphoproteomic perturbation dataset in PTMNavigator.**
**A** Edited screenshot from PTMNavigator showing an excerpt from the pathway diagram "mTOR Signaling Pathway - hsa04150" and the modified peptides significantly regulated by three of the drugs in Bekker-Jensen et al. Note that the authors added the node for EGFR. It is not part of the original pathway diagram but is known to interact with GRB2 and PI3K. The blue borders indicate kinases whose activity was significantly decreased (Score <−0.5, FDR < 0.1) according to KSEA analysis. Top: Lapatinib. Middle: Wortmannin. Bottom: Temsirolimus. **B** Edited screenshot from PTMNavigator showing an excerpt from the pathway diagram "MAPK signaling pathway - hsa04010" and the modified peptides regulated by two of the drugs in Bekker-Jensen et al. Regulated PTMs are colored by the size of the effect (Log2 Fold Change). Left: Cobimetinib. Right: Sorafenib. Source data are provided as a Source Data file.

upstream kinase in PhosphoSitePlus and PHONEMeS did not find a link, however, the similarities in regulation (Supplementary Fig. 10) suggest that it is part of the same phosphoproteomics signature. Therefore, it remains to be investigated which kinases are directly responsible for phosphorylating this site, how essential these phosphorylations are to the signaling pathway, and whether this finding translates to other biological systems. Taken together, this shows that PTMNavigator can support researchers in functionalizing

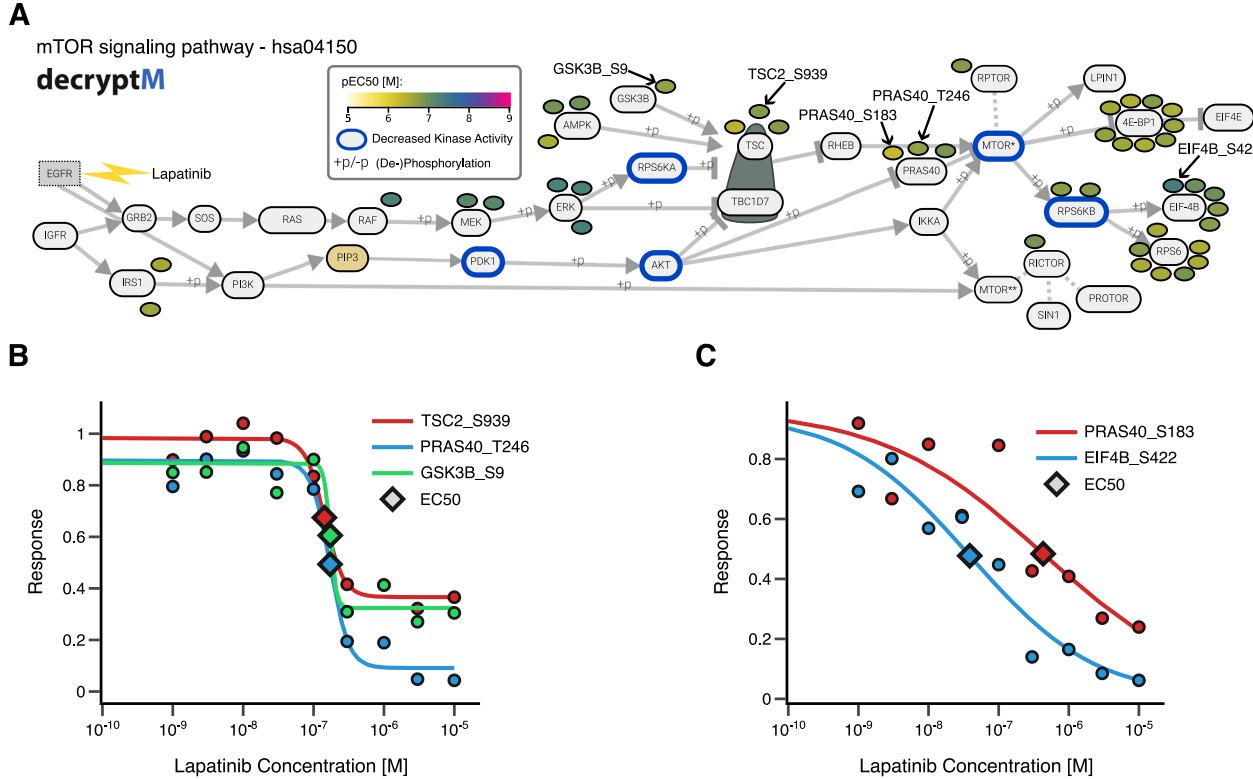

**Fig. 5 | PTMNavigator supports the analysis of drug dose-dependent perturbation data. A** Edited screenshot from PTMNavigator showing an excerpt from the pathway diagram "mTOR Signaling Pathway - hsa04150" and the modified peptides regulated in the Lapatinib experiment of Zecha et al. Note that the authors added the node for EGFR. It is not part of the original pathway diagram but is known to interact with GRB2 and PI3K. The blue borders indicate kinases whose activity was significantly decreased (Score < −0.5, FDR < 0.1) according to RoKAI refinement followed by KSEA analysis. **B** decryptM curves from the Lapatinib dataset for those peptides that were also significantly downregulated in the Bekker-Jensen data. **C** decryptM curves with the highest ($pEC_{50} = 7.41$) and lowest ($pEC_{50} = 6.37$) potency in the mTOR signaling pathway. Source data are provided as a Source Data file.

uncharacterized PTM sites and thus aid in proposing putative new links in pathways by combining prior knowledge with experimental data.

## Discussion

PTMNavigator is a tool for visualizing quantitative PTM data in the context of cellular pathways, which supports users in analyzing complex PTM proteomics data. Its current implementation in the ProteomicsDB platform allows the exploration of pathways of ten organisms using pathway annotations from KEGG and WikiPathways and functional annotations from PhosphoSitePlus. The integration of automated enrichment analyses adds another dimension of information, and the possibility of creating user-defined pathways from scratch and sharing them makes PTMNavigator versatile and unique. Other databases, such as BioCyc[48] and Reactome[8], may be utilized in future iterations of the tool to further extend the list of available pathways. We envision it will become a platform for knowledge exchange in systems biology. We consciously realized this software as a website instead of a desktop application. We believe that this makes it more accessible, as users will not have to install it or be concerned about the computational power of their desktop computers. The core visualization component of our software was built on the Web Component standards, making it easy to reuse in web interfaces outside of ProteomicsDB.

We demonstrated how PTMNavigator facilitates the interpretation of pathway enrichment analyses, makes perturbed signaling traceable, leverages the novel information contained in dose-dependent PTM data, and generates hypotheses for follow-up experiments to study unannotated modification sites. To our

knowledge, PTMNavigator is the only available software that can graphically illustrate how regulated PTMs are distributed within pathways.

PTMNavigator gives a bird's-eye view of PTMs within a pathway while simultaneously integrating multiple sources of information on particular sites. This accelerates PTM perturbation analysis considerably, especially in large-scale data sets. Furthermore, it fosters the identification of new kinase-substrate relationships and signaling pathway crosstalk, as we exemplified for the phosphoproteomic response to MEK inhibitors in A549 cells.

There are usually multiple pathways that depict a signaling cascade of interest, and choosing a single one can seem arbitrary. We want to note that for conciseness, in this manuscript we always selected a single KEGG or WikiPathways diagram for visualization of a dataset, while in practice, we recommend that the user should look at all diagrams of highly enriched pathways to get an idea of how interchangeable the visualizations are and see if specific subnetworks are shared between them. For example, the visualization of MEK inhibition, for which we chose "Regulation of actin cytoskeleton – WP51" (Fig. 6A), would, for our purpose have worked just as well with the eponymous KEGG diagram "Regulation of actin cytoskeleton – hsa04810", as it contains the same subnetwork of SOS1, RAF1, MAPK1/3, GRLF1, and MYPT1. Meanwhile, for other datasets or visualization purposes, one diagram might be preferable over the other, and it is usually impossible to decide without inspecting both. This is also the case if one of the pathways has a substantially higher relevance score – this can happen because the other pathway contains more additional proteins that "water down" the score. However, that pathway can still be the better choice for visualization if the subnetwork of interest is more appropriately represented.

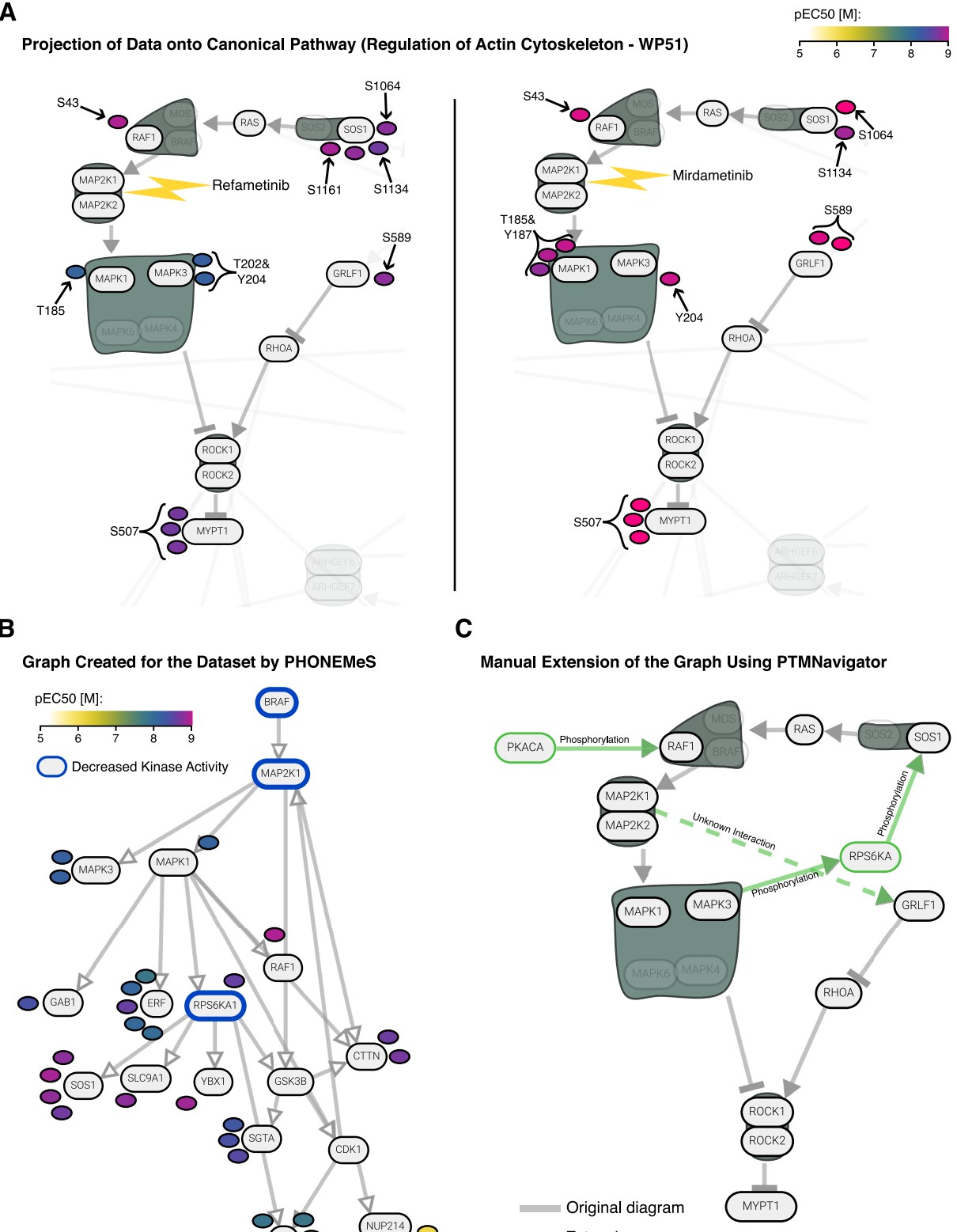

**Fig. 6 | Extending a canonical pathway with PTMNavigator. A** Edited screenshot from PTMNavigator showing an excerpt from the pathway diagram "Regulation of Actin Cytoskeleton – WP51" and the modified peptides regulated in the Refametinib (left) and Mirdametinib (right) decryptM experiments. All regulations are down-regulations. **B** Pathway graph of the Refametinib dataset generated by PHONEMeS.

The kinases with decreased activity (determined by RoKAI + KSEA) were used as perturbation targets in the PHONEMeS input. **C** Proposed manual extension to the diagram "WP51", created using the "Editing Mode" within PTMNavigator. For illustrative purposes, nodes and edges added by hand are shown in green. Source data for (A, B) are provided as a Source Data file.

Current PEAs are far from ideal for analyzing PTM data sets, and their outcomes must be taken with considerable caution. Nonetheless, PEAs currently provide the best starting point, so PTMNavigator uses PEA to pre-sort pathways by putative relevance. The initial ranking should help users prioritize pathway engagement, but users should keep in mind that some high-ranked pathways are not biologically relevant and are PEA artifacts. This is often rooted in pathway annotations being highly redundant (i.e., the same well-studied proteins show up in many well-studied pathways). On the other hand, pathways with non-significant scores are often worth examining, and additional resources, such as previous literature, should be considered when selecting diagrams for visualization. If better pathway ranking procedures are developed in the future, PTMNavigator can easily be updated to use those. We emphasize that PTMNavigator is a visualization tool and that interpreting the results remains the user's responsibility. While we believe the current work constitutes a substantial advance, we point out several caveats: i) Differential or dose-response analysis is not part of PTMNavigator, and users must perform this before data upload. PTMNavigator only shows what its users already deemed significantly regulated. This is also a benefit, as it makes PTMNavigator flexible to many experimental settings. ii) The currently available canonical pathway diagrams are heterogeneous in structure, and, for PTM research, they may sometimes even be misleading. One example is the phosphorylation and dephosphorylation annotations in KEGG diagrams ("±p"). A phosphorylating connection from kinase A to protein B means that at least one site on B is a substrate of A, not necessarily (actually not likely) all of them. Still, for a user, the latter may seem implied. Furthermore, we could only retain those details of the pathway diagram that were also represented in the KGML or GPML files. Some of the pathways resemble drawings rather than formal diagrams, and the relationships between pathway members are not encoded as edges in the graph (consider, for example, the KEGG pathway "Nucleocytoplasmic transport – hsa03013" and the WikiPathways *D. melanogaster* pathway 'Homologous recombination – WP1205'). In PTMNavigator, these pathways contain few or no edges and can only serve as a collection of functionally related proteins. iii) Pathway definitions are typically generalized from a large amount of heterogeneous data. Consequently, they cannot be specific for a particular tissue or cell type, and some proteins depicted in a pathway might not be expressed or active in the cells in which an experiment was performed. iv) A single node may represent a single gene or gene product. Still, it can also summarize a subfamily of proteins (such as MAPK) or even an entire class of proteins (such as all receptor tyrosine kinases). Conversely, a "group" node can represent a protein complex and a set of alternatives for a step in the pathway. Users of the pathway diagram need to be wary of these disparate definitions. v) The current knowledge of pathways is limited, and many relationships are not yet discovered. PTMNavigator can only show what is already known, and its use for discovering novel signaling cascades or nodes is necessarily limited. As a consequence, PTMNavigator will become more potent as pathway databases become more comprehensive in the future.

## Methods

### Creating a pathway database in ProteomicsDB

We assembled a collection of pathways from ten species within ProteomicsDB based on the reference pathway databases in KEGG (Release 107.0, 1 July 2023) and WikiPathways (Release 20230610, 6 June 2023). Both projects provide textual pathway definitions in variations of the extensible markup language (XML) format (KEGG: KEGG Markup Language (KGML); WikiPathways: Graphical Pathway Markup Language (GPML)), which we used as a starting point. There are differences in the detailed pathway specifications of KEGG and WikiPathways (see the following two subsections). Still, essentially, both define a pathway as a mixed graph consisting of a set of nodes and a set of edges (both directed and undirected) that connect the nodes. There are two classes of nodes: *regular nodes* have a description that includes 2-dimensional cartesian coordinates as well as information on the entity the node represents (for example, a gene name), and *group nodes* do not contain any additional description. Instead, they are defined solely by a set of regular nodes contained in them. We wrote a Python package termed *pathway-importer* that converts both types of pathways into JavaScript Object Notation (JSON) format. The JSON format is better suited for use with a web application since it can be parsed at runtime without further conversion (in contrast to XML). We did not import pathways that contained zero gene or gene product nodes since they are not interesting for the study of cellular signaling (this includes, for example, metabolic pathways that are focused only on metabolites).

### Preprocessing of KEGG pathway diagrams

We retrieved the list of pathways for each organism and downloaded the KGML files using the KEGG REST API (https://rest.kegg.jp). KGML files distinguish four types of nodes: *gene, compound, ortholog*, and *map*. To reduce the complexity of our internal representation, we summarized the *gene* and *ortholog* nodes into a common *gene_protein* node type. *map* nodes were renamed as *pathway* nodes since they describe connections to other pathways and *compound* nodes were not modified (Supplementary Fig. 2). In KGML files, nodes are labeled using KEGG identifiers, which we mapped to UniProt accession numbers using the "idmapping" service of the UniProt REST API (as described here: https://www.uniprot.org/help/id_mapping). To maximize the number of nodes that had a human-readable name, we subsequently mapped the UniProt accession numbers to gene names (also using the UniProt API).

KEGG pathways have two classes of edges: *relations*, which connect exactly two nodes, and *reactions*, which can also represent many-to-many relationships by specifying a list of substrates and a list of products. We intended to represent only one-to-one relationships in ProteomicsDB. Therefore, we replaced each reaction with an equivalent set of *relations*, connecting each substrate-product pair by an individual edge. KEGG *relations* can have 19 different subtypes, which we

retained in our JSON representation with two exceptions (Supplementary Fig. 3):

a) The types "indirect" and "indirect effect" were summarized into one type, "indirect."
b) "inhibition" and "repression" were summarized into one type, "inhibition."

### Preprocessing of WikiPathways pathway diagrams

We downloaded GPML files from http://data.wikipathways.org/current/gpml. These pathways, maintained by a large, open community, are less standardized than in KEGG. The pathway collection we downloaded distinguishes 13 types of nodes, which we mapped to the same types as the nodes from the KEGG pathways. In addition to the *gene_protein*, *compound*, and *pathway* types, we introduced a fourth type *misc* to account for WikiPathways nodes that could not be assigned one of the other categories (Supplementary Fig. 2). In GPML files, each node is annotated with a reference to an external database, such as Entrez or Ensembl. To unify these diverse annotations, we mapped them to UniProt Accession Numbers where possible, using the UniProt API (Supplementary Fig. 4). Edges in GPML files have 24 different types. Ten of these were converted into one of the KEGG edge types, and the other 14 were retained as additional types (Supplementary Fig. 3). A specific feature of the GPML specification is that edges may end on another edge instead of a node. This is represented by an additional tag in the XML schema called *anchor*. To keep our

representation simple, we replaced *anchor* tags with edges that had other edges as their start or end points.

## Implementation of the *pathwaygraph* Web Component

We developed a Web Component (www.webcomponents.org), termed *pathwaygraph*, for the dynamic rendering of pathway diagrams and peptide-level experimental data. *pathwaygraph* was implemented using the Lit library (https://lit.dev/), Version 2.0.2. Our Web Component expects two inputs: A pathway diagram in JSON format (we call this "pathway skeleton") and a list of PTMs, each annotated with a regulation type (possible values are "up," "down," and "not"). Optionally, a list of proteins and their regulation types can also be supplied.

*pathwaygraph* maps each entry of the PTM list to nodes of the pathway skeleton by comparing the gene names and/or UniProt accession numbers associated with the nodes. For each matched PTM, an additional node is added to the graph definition, and an edge is added between the new node and its associated node in the skeleton. A PTM can be matched to multiple skeleton nodes, resulting in multiple PTM nodes for the same entry of the PTM list. This can happen, for example, when a protein appears twice in the same pathway because the creators wanted to describe two different routes of a signaling cascade.

After completing the matching process, *pathwaygraph* renders the graph as an SVG object using the software library D3.js[49] (Version 6.2.0). The cartesian coordinates of the pathway skeleton nodes are used for the initial layout. The edges are drawn to connect their two adjacent nodes in a straight line. When a node changes its position (e.g. by a user dragging it around), the endpoints of the edge are automatically updated. The PTM nodes, which do not have cartesian coordinates assigned, are positioned using simulated physical forces (using the d3-force package (https://github.com/d3/d3-force), Version 2.1.1), which cause them to remain close to their reference skeleton node without colliding with one another. PTM nodes can be represented individually or as a single summary node for each regulation type that only displays the number of regulations (step 6 in Fig. 2A; users can toggle between the two representations). Optionally, a label consisting of the modified residue and position within the protein can be displayed above the PTM node.

The different node types (*gene_protein*, *compound*, *pathway*) are distinguished by different colors. Group nodes are represented as polygons around the convex hull of all members of the group. Four types of edges are distinguished visually by different dash patterns or arrowheads: *activation*, *inhibition*, *binding/association*, and *indirect*. All remaining edge subtypes are rendered using the same style. Only edges from the skeleton are rendered. The edges between PTM and skeleton nodes are invisible (but their presence is required for the force simulation).

There are three possible color schemes for the PTM nodes: the default and most straightforward option is to color them by their regulation type ("up" nodes are colored red, "down" nodes blue, and "not" nodes gray). Alternatively, nodes can be colored by fold change or (in the case of decryptM data) by potency (using EC50 values estimated from the dose–response curve fit). In both cases, the full range of values is determined after mapping the PTM nodes to the skeleton. A continuous color gradient is then calculated between the minimum and the maximum value and applied to each node.

Each PTM in the input can be annotated with arbitrary details. In the case of phosphoproteomics data, these details may include upstream kinases. Suppose such annotations are present, and the kinases are also present in the currently displayed pathway skeleton. In that case, *pathwaygraph* will automatically add the kinase-substrate relationships as additional edges to the diagram that can be shown during the analysis.

In addition to PTM-level and protein-level data, a list of perturbation targets (e.g. estimated by a KSEA) may be supplied. Blue and red circles around the respective nodes can optionally highlight these targets within the pathway diagram.

In addition to pathway viewing and projection of PTM datasets, we implemented a second application mode for *pathwaygraph*, in which pathway diagrams can be edited. This mode allows the addition, removal, and rearrangement of nodes, node groups, and edges. Details of nodes and edges (such as labels or gene identifiers) can also be edited. Diagrams can be exported in the same JSON format as the canonical "pathway skeletons" and stored in ProteomicsDB. It is also possible to upload JSON files from a user's local machine for editing and storing in ProteomicsDB.

Using the package @api-client/context-menu, v0.4.1, we developed a context menu for *pathwaygraph* which allows the user to interact with the application. The context menu is dynamic, and its contents depend on the current state of the application (e.g., it shows different options for viewing and editing).

## Implementation of PTMNavigator

We implemented PTMNavigator as a Vue.js component (https://vuejs.org). It is essentially a wrapper around *pathwaygraph* that facilitates interaction of the Web Component with the user and the backend of ProteomicsDB and can provide additional information, such as the results of enrichment analyses (see below). PTMNavigator handles user interactions such as selecting datasets, organisms, and pathways or applying filters on the loaded data by calling the necessary endpoints of the ProteomicsDB API, processing the returned data, and passing it on to *pathwaygraph*. Custom pathway diagrams can also be saved to and restored from ProteomicsDB (associated with the user's Session ID) and downloaded and uploaded in JSON format.

PTMNavigator is embedded into ProteomicsDB, which is also Vue.js-based since its latest release[24]. The tool can be found in the "Analytics" section of ProteomicsDB: https://www.proteomicsdb.org/analytics/ptmNavigator.

## Implementation of a service for enrichment analyses

Using the *flask* framework (version 2.0.3), we implemented a Python-based web server that provides several common enrichment analysis algorithms for proteomics data, with a focus on phosphoproteomics. An Application Programming Interface (API) allows applications and human users to send POST requests to the server. Since each method that the service offers requires a different way of structuring the input data, we implemented preprocessing scripts that convert JSON input into the required formats so that the input the user has to provide is as similar as possible for all algorithms. Each algorithm's result is then converted into JSON and sent back to the requester. We mostly used existing Python or R packages, except where noted. The service is publicly available under this web address: https://enrichment.kusterlab.org/main_enrichment-server/. The code can be accessed here: https://github.com/kusterlab/enrichment-server. In the following, we describe the methods offered by our service at the time of writing. For details on the input formats, please refer to the GitHub repository, where we provide examples for every endpoint.

**PTM signature enrichment analysis (PTM-SEA).** We use the R package ssGSEA2.0 (https://github.com/broadinstitute/ssGSEA2.0), commit hash 4b5198f (20 July 2023). As a database, we use PTMSigDB v2.0.0, "flanking" and "uniprot". Expected input is a list of PTM sites and a numeric value for each experiment (e.g. normalized measured expressions). Duplicates are eliminated during preprocessing by retaining the entry with the maximum absolute expression value for each site. The algorithm is invoked with the export (-e) parameter set to FALSE and all other parameters set to their default values. As the test statistic, we use the difference between the Empirical Cumulative

Distribution Function (ECDF) of peptide ranks in a signature and the ECDF of the remaining peptides' ranks (this is the default setting for the ssGSEA2.0 algorithm; for details, please refer to the original publication). From the resulting .gct file, the "Signature ID" column and the "Overlap", "adj p-val", and "Score" columns for every processed experiment are returned.

**Gene-centric and gene-centric redundant single sample gene set enrichment analysis.** Again, we use the R package ssGSEA2.0. As a database, we use the subset of all KEGG and WikiPathways signatures from MSigDB. To match the pathways available in PTMNavigator, we excluded "KEGG medicus" signatures. The expected input is a list of gene symbols and their expression in each experiment. In the case of Gene-Centric analysis, duplicates are eliminated as described for PTM-SEA. In the case of Gene-Centric Redundant analysis, duplicates are not eliminated. The same test statistic as for PTM-SEA is applied (difference between ECDFs), and the same columns as for PTM-SEA are returned.

**Kinase-Substrate Enrichment Analysis (KSEA).** We use the *kinact* Python package (https://github.com/saezlab/kinact, commit hash 572d3a8) which performs KSEA as described by Casado et al. by comparing the mean fold change among the set of substrates of a kinase to an expected value. As prior knowledge, we use the set of kinase-substrate relationships from PhosphoSitePlus, retrieved using OmniPath (https://pypath.omnipathdb.org/) on 2 February 2024, and converted into an adjacency matrix by a custom Python script. Expected input is a list of phosphorylation sites and a numeric value for every experiment. Duplicates are eliminated as described for PTM-SEA. For each experiment, scores and adjusted $p$ values are returned.

**Robust Kinase Activity Inference (RoKAI)+KSEA.** We use the R package RokaiApp (https://github.com/serhan-yilmaz/RokaiApp), commit hash e2eb10a. As prior knowledge, we use the file "rokai_network_data_uniprotkb_human.rds" provided by the RokaiApp repository. The expected user input is the same as for KSEA. In addition to RoKAI's default Kinase-Substrate network, all optional components of the RoKAI circuit system are used (PPI, SD, CoEv). The resulting refined phosphorylation profiles are subsequently processed by KSEA as described above.

**PHONEMeS.** We use the R package PHONEMeS (https://github.com/saezlab/PHONEMeS), v2.0.1, as well as Cytoscape v3.10.1 equipped with the yFiles plugin (https://www.yworks.com/products/yfiles-layout-algorithms-for-cytoscape), v1.1.4. As prior knowledge network, we use the concatenation of the "phonemesPKN" and "phonemesKSN" networks, which describe ground truth protein-protein interactions and kinase-substrate relationships. The expected user input is, for each experiment, a list of up- and down-regulated perturbation targets (in gene symbol format) and a list of sites with numeric values (e.g., normalized measured expression, Fold Change, or *t*-statistic). The PHONEMeS algorithm is then invoked for each experiment with n_steps_pruning set to 2 and default values for all other parameters, and using the IBM cplex optimizer (https://www.ibm.com/products/ilog-cplex-optimization-studio/cplex-optimizer) as solver for the Integer Linear Program. If no solution is found, the algorithm is rerun with n_steps_pruning set to 3. Then, the resulting network for each experiment is converted into a *pathwaygraph* pathway skeleton: first, all nodes representing a phosphorylation site are replaced by a node for the protein to which the phosphorylation site corresponds. Then, duplicated *node-edge-node* triplets and self-loops are removed. Next, the network is loaded into Cytoscape (we use the Python package py4cytoscape, v1.9.0., for programmatic interaction with Cytoscape). The yFiles "Hierarchic Layout" is applied, and the graph is exported as a

".cx" file. Using custom Python code, this file is converted into a JSON file with the 'pathway skeleton' format that can be parsed by *pathwaygraph*. The JSON file is returned to the requester.

**Motif enrichment.** We use in-house Python code that essentially reimplements the service provided here: https://kinase-library.phosphosite.org/ea. In brief, from the kinase library data, we created position-specific scoring matrices for each kinase that reflect the preference for every amino acid to appear at every position of the kinase's substrate motif. We scored every phosphopeptide in PhosphoSitePlus against the matrix of each kinase to obtain a background score distribution. The expected user input is then a list of phosphopeptide sequences, the UniProt Accession Numbers of the corresponding proteins, and, for each experiment, the regulation category of this peptide ('up'/'down'/'not'). Every peptide from the user input is scored against every kinase, and the result is compared to the background to obtain quantiles. The 15 highest-scoring kinases for every site are determined. Then, each kinase that appears in the result is tested for significant overrepresentation through a Fisher's exact test (including multiple testing corrections using the Benjamini-Hochberg method). The test results for each kinase in each experiment are converted into JSON format and returned to the requester.

**KEA3.** We use the application programming interface (API) of KEA3. The expected user input is a list of proteins (gene symbols) for each experiment. The enrichment service sends a POST request to the URL https://amp.pharm.mssm.edu/kea3/api/enrich/. The tables "Integrated--meanRank" and "Integrated--topRank" are extracted from the response and returned to the requester.

**KSTAR.** We use the kstar Python package, v0.5.3. The expected user input is a list of phosphopeptide sequences, the associated UniProt Accession Numbers, and a numeric value for each peptide in each experiment. The peptides are converted into "flanking sequence" format (7 adjacent amino acids in both directions) and annotated with their position within the protein using the psite_annotation package (v.0.5.0) and the reference proteome from PhosphoSitePlus (Version from 2 November 2020). KSTAR is then executed using "mean" as the aggregation method and 0 as the threshold (which means that values are regarded as significant, therefore, non-significant regulations need to be filtered out by the requester in advance). The algorithm is run twice, first with "greater" set to FALSE, then TRUE, to capture over- and underactive kinases. This is executed for both modification types ("ST" and "Y"). The $p$-values of the hypergeometric test are log-transformed and, in the "greater = =TRUE" run, multiplied by $-1$. This ensures activated kinases receive a score $>0$, while inactivated kinases get a score $<0$. These scores are returned to the requester. Note that the KSTAR protocol recommends conducting analyses on randomized datasets afterward and conducting Mann-Whitney $U$-tests to obtain more robust $p$-values. Since this time-intensive calculation requires significantly more memory than the other enrichment algorithms, we decided against including this step in our enrichment service.

### Processing of user data upload

We extended the upload page of ProteomicsDB (www.proteomicsdb.org/analytics/customDataUpload) to allow the upload of data for PTMNavigator in CSV format. It is possible to upload conventional quantitative proteomics ("Fold Change") data as well as decryptM data on both PTM and protein levels. In the case of phosphorylation data, users can upload data on both peptide and p-site level; for other PTMs, only peptide-level upload is possible at this time. Example files for all input types are provided for download on the website (see also

Supplementary Data 1). In the case of decryptM data, the "curves.txt" file output by the decryptM pipeline or CurveCurator can directly be uploaded together with the TOML file containing the experimental parameters. The uploaded data is stored temporarily in ProteomicsDB and is deleted if not accessed for 14 days. After completing the upload, requests are sent to the endpoints of the enrichment service (see above). For phosphoproteomics data, all endpoints are queried. For other PTM and protein-level data, only the gene-centric and gene-centric redundant GSEA are performed. In the case of the PHONEMeS +Cytoscape endpoint, perturbation targets are required as input in addition to the phosphoproteomic data. Here, the results of the RoKAI +KSEA analysis are used. The three highest-scoring kinases for each experiment are regarded as perturbation targets. The pathways returned by the endpoint are added to the user's custom pathway diagram list. All other enrichment results are inserted into a dedicated table in ProteomicsDB. To limit access to the data, the user receives a personalized 32-digit alphanumeric session identifier during the upload that needs to be entered in PTMNavigator to retrieve the uploaded data.

When a user visits https://proteomicsdb.org/analytics/ptm Navigator, the custom pathways associated with their Session ID are automatically retrieved. When the user requests a dataset they had previously uploaded, all enrichment analysis results for this dataset are also retrieved. The canonical KEGG and WikiPathways pathways are sorted by gene-centric redundant GSEA score.

### Highlighting of perturbed kinases in the pathway graph

*pathwaygraph* can highlight protein nodes that have been deemed significantly perturbed using red (up), blue (down), or purple (unknown direction) circles. A dedicated subcomponent within PTMNavigator allows the user to control this feature for over- or under-activated kinases in phosphoproteomics data. When loading one or multiple datasets, the component receives as input the results of all kinase activity inference analyses that were carried out for the data (KSEA, RoKAI+KSEA, Motif Enrichment, KEA3, KSTAR). Users can decide which method to use for highlighting and filtering the data by significance and/or score (depending on the metrics the respective method uses). The filtered kinases are forwarded to *pathwaygraph*, and if they appear in a chosen pathway diagram, they will automatically be highlighted.

### Import of PhosphoSitePlus annotations and functional scores

We downloaded the files "Regulatory_sites.gz" and "Kinase_Substrate_Dataset.gz" from PhosphoSitePlus (https://www.phosphosite.org/staticDownloads; accessed 18 October 2023). Using a custom Python script, we mapped all annotations to the in silico digested human reference proteome in ProteomicsDB and created API endpoints to make the information accessible from the PTMNavigator frontend. The phosphorylation site functional scores were imported analogously, using Supplementary Table 3 from the paper by Ochoa and colleagues[33]. During retrieval of a dataset, the associated annotations are collected from ProteomicsDB and forwarded to PTMNavigator alongside the data. This allows, for example, to display kinase-substrate annotations from PhosphoSitePlus as additional edges within the diagram.

### Preprocessing of data from Bekker-Jensen et al.

Please refer to the original publication for detailed experimental procedures of the kinase inhibitor screen. Briefly, RPE1 cells were treated with either 0.1 µM or 1 µM of inhibitor for 30 min, followed by stimulation with EGF for 10 min. All experiments were performed in triplicates, except for an EGF-stimulation-only control, which was performed in duplicates. After extraction of proteins, digestion into peptides, and Ti-IMAC-based phosphopeptide enrichment, the samples were measured using LC-MS/MS in Data-Independent Acquisition

(DIA) mode. The data was then processed using Spectronaut to identify and quantify phosphorylated peptides.

Our analysis of the data differed from that of Bekker-Jensen et al. in that they performed multiple sample significance testing (ANOVA), comparing all conditions against the control together. In contrast, we intended to compare each condition to the control separately using *t*-tests. We downloaded the Spectronaut output file from ProteomeXChange using the Identifier PXD014525 ("20190416_214744_Kinase-inhibitor-screen-3reps-STY-version13-LocProb-075-Score-1_Report.txt") and imported it into Perseus (version 1.6.2.3). We transformed the normalized intensities into phosphorylation site tables using the Peptide Collapse plugin (https://github.com/AlexHgO/Perseus_Plugin_Peptide_Collapse), specifying EG.PTMAssayProbability as grouping column. We filtered for three valid values in at least one treatment group. We log-transformed and replaced missing values by sampling from a normal distribution (setting the parameters "width" to 0.3, "downshift" to 1.8, and "mode" to "Total matrix"). We then performed two-sided *t*-tests between the 60 treatment conditions (30 drugs, 2 doses) and the control condition for each quantified peptide. *p* values were corrected for multiple testing using the Benjamini-Hochberg procedure (FDR = 10%). Finally, we summarized the data into 30 conditions (one per inhibitor) and eliminated duplicated peptides using the following rules:

I. If the peptide appeared multiple times in the same experiment (which can happen for multiply phosphorylated peptides), retain only the most significant entry (lowest adjusted *p*-value).
II. If a peptide was neither significantly regulated in the low-dose experiment nor the high-dose experiment of a compound, retain only the entry from the high-dose experiment.
III. If a peptide was significantly regulated in the high-dose experiment, retain only this entry.
IV. If a peptide was significantly regulated only in the low-dose experiment, retain only this entry.

The resulting tables were formatted to fulfill the input criteria of PTMNavigator.

Pathway enrichment was performed as described above.

The code of this analysis (starting from the Perseus output and ending at the creation of PTMNavigator input files) is available as a Jupyter Notebook (see "Data Availability").

### Preprocessing of data from Zecha et al.

For details on the decryptM experimental protocol, please refer to the supplementary methods in the original publication. In brief: In the Lapatinib experiment, MDA-MB-175-VII cells (human breast cancer) were treated with a series of Lapatinib for 2 h (concentrations: 1, 3, 10, 30, 100, 300, 1000, 3000, 10,000 nM) or a DMSO-only control. In the Refametinib/Mirdametinib experiments, human lung adenocarcinoma A549 cells were treated with a series of Mirdametinib (PD325901) or Refametinib concentrations for 60 minutes (concentrations: 1, 3, 10, 30, 100, 300, 1000, 3000, 10,000 nM), or a DMSO-only control. In the Dasatinib experiments, A431 epidermoid carcinoma epithelial cells, K562 lymphoblast cells, and A549 cells were treated with Dasatinib (concentrations: 0.03, 0.3, 1, 3, 10, 30, 100, 300, 1000, 10,000 nM for A431 cells; 1, 3, 10, 30, 100, 300, 1000, 3000, 10,000 nM for A549 cells; 0.01, 0.03, 0.1, 0.3, 1, 3, 10, 30, 100 nM for K562 cells) or a DMSO-only control for 30 min (60 min for A549 cells). Proteins were extracted from the cells and digested using Trypsin. The samples were multiplexed using tandem mass tags (TMT). Phosphopeptide enrichment was performed using Fe-IMAC, after which the samples were separated into six fractions using bRP-fractionation to increase phosphoproteomic coverage. The fractions were then measured separately using LC-MS/MS in data-dependent acquisition (DDA) mode. TMT

phosphopeptides were identified and quantified using MaxQuant (version 1.6.12.0).

We downloaded the following archives containing MaxQuant output from ProteomeXChange using the identifier PXD037285:

a. Lapatinib experiment: HER2_Inhibitors_Phosphoproteome_dd.zip
b. Refametinib experiment: 10_Kinase_Inhibitors_Rep_Phosphoproteome.zip
c. Mirdametinib experiment: 10_Kinase_Inhibitors_2KI_Phosphoproteome.zip
d. Dasatinib A431 experiment: 3_EGFR_Inhibitors_Phosphoproteome.zip
e. Dasatinib A549 experiment:10_Kinase_Inhibitors_6KI_Phosphoproteome.zip
f. Dasatinib K562 experiment: Dasatinib_Triplicates_Phosphoproteome_MS3.zip

We filtered the evidence.txt files for rows containing the experiments of interest. The filtered files, together with TOML parameter files which we constructed manually, were used as input for CurveCurator (https://github.com/kusterlab/curve_curator). Using CurveCurator, we fitted 4-parameter log-logistic dose-response curves for every quantified peptide and obtained significance estimates for the regulations. Then, we imported the curves file and the TOML into PTMNavigator. For the Refametinib and Mirdametinib datasets, pathway enrichment was performed as described above.

### Preprocessing of data from Lee et al.

We downloaded the Datasets EV4 (Full Proteome Data) and EV5 (Phosphoproteome Data) from here: https://doi.org/10.1038/s44320-023-00004-7 and used the sheets containing log-transformed and median-centric normalized iBAQ Values with imputations ("normalized_FP_log10_imput_hmin" and "normalized_PP_log10_imput_hmin", respectively). We calculated the difference in iBAQ log fold changes between the A204 and RD-ES cell lines. As a threshold for regulation, we set an iBAQ log fold change difference of three for the full proteome data and four for the phosphoproteomic data (i.e., peptides/proteins higher in A204 are annotated as "up", whereas peptides/proteins higher in RD-ES are annotated as "down"). The resulting files were loaded into PTMNavigator.

### Reporting summary

Further information on research design is available in the Nature Portfolio Reporting Summary linked to this article.

## Data availability

Experimental data were downloaded from ProteomeXChange as described in the Methods, using the identifiers PXD014525 (Bekker-Jensen et al.) and PXD037285 (Zecha et al.), as well as from the website of Molecular Systems Biology: https://doi.org/10.1038/s44320-023-00004-7 (Lee et al.). The following files are available on Zenodo (https://doi.org/10.5281/zenodo.12720528): For the Bekker-Jensen data: Perseus output (log-transformed phosphorylation site intensities) (.txt); A Jupyter Notebook describing how to convert the Perseus output into PTMNavigator input (ipynb); PTMNavigator input (.xlsx); Enrichment analysis results: GCR, PTM-SEA, KSEA, RoKAI+KSEA (all.xlsx); For the Zecha data: CurveCurator parameter files (.toml); CurveCurator output (.xlsx); Enrichment analysis results: GCR, PTM-SEA, KSEA, RoKAI+KSEA (all .xlsx); For the Lee data: PTMNavigator input (.xlsx). All other data are available in the main text or the supplementary materials. Source data are provided with this paper.

## Code availability

The code to import and preprocess the pathway diagrams is available at https://github.com/kusterlab/pathway-importer, as well as on

Zenodo[50]. *pathwaygraph* is part of the series of Bio-Web Components developed by our lab, @biowc, and is available as a JavaScript module here: www.npmjs.com/package/@biowc/pathwaygraph. The code can be accessed here: www.github.com/kusterlab/biowc-pathwaygraph, as well as on Zenodo[51]. The source code of PTMNavigator can be accessed here: https://github.com/wilhelm-lab/proteomicsdb-vue/blob/main/src/views/analytics/PTMNavigator.vue. The source code of the Enrichment Server can be accessed here: www.github.com/kusterlab/enrichment-server, as well as on Zenodo[52].

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

## Acknowledgements

We thank Chien-Yun Lee and Stefanie Höfer for testing the software and providing valuable feedback. We thank all other members of the Kuster, Lee, and Wilhelm Labs, as well as the Saez Lab, for fruitful discussions. Parts of Fig. 1 were created with BioRender.com. This work was partly funded by the European Research Council (ERC Advanced Grant No. 833710; J.M., B.K., F.B., and M.T.), and the German Federal Ministry of Education and Research (DIAS, FKZ 031L0168; J.M. & DROP2AI, FKZ 031L0305A; M.W., M.T., and B.K.).

## Author contributions

Conceptualization: M.T., F.B., J.M. Software: J.M., M.S. Data Analysis: J.M., M.T., F.B. Writing – original draft: J.M. Writing – review & editing: M.T., B.K., F.B., M.W., J.M., M.S. Supervision & Project Administration: M.T., M.W., and B.K.

## Funding

## Competing interests

B.K. and M.W. are founders and shareholders of OmicScouts and MSAID. They have no operational role in either company. All other authors declare that they have no competing interests.
