## [Transparent Peer Review file · Nature Communications]

PTMNavigator: Interactive Visualization of Differentially Regulated Post-Translational Modifications in Cellular Signaling Pathways

Corresponding Author: Dr Matthew The

Version 0:

Reviewer comments:

Reviewer #1

(Remarks to the Author)

Manuscript review points

This manuscript describes PTMNavigator which as a new user-friendly data analysis tool to visualize quantitative phosphorylation data on signaling pathway maps. The article is clear and well written. The PTMNavigator tool will certainly be helpful to many users. However, it is not clear to me that this manuscript presents a significant enough advancement in the field to warrant publication in this journal. The manuscript at present is better suited for another journal. There are many desired features that would need to be added to the tool to make this resource a significant advancement, many of these features are addressed by the authors themselves in the discussion section. It is clear in the manuscript that extensive manual interpretation is required to draw conclusions from the pathway maps. In many cases this data could already be scored and quantified by the tool in order to help users reach biological interpretations faster and easier. Some specific comments:

1. The authors point out the challenge of interpreting pathway enrichments that are not biologically relevant for the samples analysed. These pathways are enriched based on a small subset of proteins (or subnetwork) that is actually a shared feature with other, more relevant pathways. Why not provided weighted protein-centric scoring to identify these subnetworks or protein subsets that are shared across multiple pathways? An algorithm can surely identify and visualize these overlaps better than manual curation of the different pathway hits. Automating the identification of such shared subnetworks will identify protein drivers embedded within larger pathways and will help avoid misleading biological insight from irrelevant pathway hits.
2. Pathway expansions performed in Fig 5D is helpful and should already be a feature in this tool. If it is not possible to expand the visual, at least the metadata associated with the sites should be added to the existing pathway maps.
3. Can the extent signlaing crosstalk between pathways be quantified?
4. Kinase activity inference should be a feature in this tool. It is a helpful readout which can already been found in tools like PTMgsea. Connecting evidence of kinase activation with different pathway signaling nodes in a visualisation would be a unique contribution from this tool.
5. Phosphorylation site values go up or down, but what about protein activity? Are these increase or decreases in phosphorylation associated with activation or deactivation of protein function? Certainly this info is only known for a small subset of proteins, but it should be accessible and annotated for those sites. Ultimately this would be a biological interpretation that is sought after in quantitative phosphoproteomics data.
6. Integration of a feature that allows the magnitude of phosphorylation changes to be highlighted over proteome changes for any given protein would be a very interesting. You could visualize which sites are trending with protein change and which have dynamic regulation beyond protein level changes.
7. There are already other tool that can visualize phosphphorylation changes within protein networks, such as PhosphoPath within Cytoscape.

Reviewer #2

(Remarks to the Author)

Summary:

PTMNavigator offers a novel and user friendly interface to map individual post-translational modifications with fold change or drug potency values onto proteins along pathway diagrams imported from KEGG and wikipathways with links to functional annotations from PhosphositePlus and from Ochoa et al (2020). The tool includes the most pathway diagrams of any tool to date and best visual presentation of sites along a pathway when compared to Cytoscape (PTMOracle, Phosphopath), Phosphomatics, KEGGViewer, Wikipathways and PhosphositePlus. The paper successfully demonstrated the visualization capabilities of the tool with internal and external datasets. The added benefit to the field is exemplified in Supplementary Figure 4, which nicely illustrates how signaling differs between cell lines following dasatinib treatment.

Issues with the relevance scoring were discovered during analysis of the test data and were only partially acknowledged. Relevance scoring issues should be explicitly addressed in the revision as suggested below to mitigate overreliance on the ranking functionality and acknowledge previous publications outlining issues of gene-centric ranking models for PTM analysis.

As it stands, most users who are not signaling experts and study mouse or human phosphorylation pathways will benefit from starting with alternative tool PTM-SEA (Krug et al MCP, 2019) for pathway identification, followed by visualization in PTMNavigator. However, domain experts and users studying the eight other organisms uniquely represented in PTMNavigator will benefit from PTMNavigator visualization in concert with literature searching for pathway identification. Overall, this work adds a useful visualization tool to the fields of PTM signaling and systems biology.

Major comments:

A critical issue with the paper is the partial presentation of shortcomings of the gene-centric relevance ranking model imported from g:profiler and missing acknowledgement of Krug et al MCP 2019 for previous presentation of ranking issues and identification of two better ranking algorithms.

First, the paper suggests manual inspection of middle to high ranking pathways will contain the truly perturbed pathway amidst PEA artifacts (Lines 96 - 99; 317-320). However, this is not always the case, as exemplified in Temsirolimus, which targets mTOR late in the pathway and shows almost 0 relevance (Lines 174 - 189 and Fig 2A). The paper should explicitly state this issue to point readers toward inspection of all pathways despite relevance ranking.

Second, the paper suggests the KEGG and wikipathways are interchangeable, however the relevance rankings contradict this claim. The paper should explain to readers why this may be the case.

Third, Krug et al MCP (2019) outlined inferior performance of gene-centric approaches for pathway scoring relative to either gene-centric-redundant or PTM signature based scoring. Despite previously published knowledge, PTMNavigator implemented the inferior gene-centric scoring with g:profiler and arrived at similar conclusions without citing the previous findings. Further, in lines 72-73, the paper indicated aggregation of PTMs to protein level modifies pathway enrichment analysis for PTM signaling, but the variable functionality of PTMs questions the validity of this approach. Please add citations supporting the questionable impact of such an approach. Otherwise, such a claim distorts prior literature showing superior performance of gene-set-redundant scoring relative to gene-centric scoring (Krug et al. MCP (2019)).

Fourth, the paper states in lines 322-323 that PTMNavigator will be updated with better scoring models once available and in line 317 that PEAs provide currently the best starting point for pathway relevance sorting. Better scoring models are available and should be incorporated, or otherwise modify the statement. Either the gene-centric-redundant or PTM-signature-set scoring outperform gene-centric scoring per Krug et al MCP (2020). The tool will be significantly improved if the PTM-signature sets from PTM-SEA along with DeCryptM Kinase inhibitor signatures are imported into PTMNavigator and a PTM-signature score implemented. Given these implementations may require significant effort, an alternative option is to modify g:profiler implementation to test a gene-centric-redundant model as tested in Krug et al. MCP (2020); or modify statements in lines 317, 322-323.

Minor comments:

1. line 89 claims "quick" examination of pathway engagement, which is not necessarily the case if extensive manual investigation is still required. I suggest changing "quickly" to "visually."

2. Lines 96 - 99: Change to "allow users to visually inspect both high and low ranked pathways." Lines 174 - 189 and Fig 2A indicate Temsirolimus targets mTOR pathway however the mTOR pathway receives a 0 or near insignificant relevance score. Therefore, users should be aware that even the lowest ranking pathways may represent biological truth.

3. Lines 318 - 320: The paper notes that initial relevance rankings help users prioritize pathway engagement, but that some high-ranking pathways are not biologically relevant and are instead a PEA artifact. This statement should be extended to additionally indicate some 'low-ranking or unranked pathways may instead be of true biological relevance' in light of the observations above.

4. Figure 2A (lines 624-625) and lines 312 to 315 contradict each other regarding the similarity between KEGG and Wikipathways. Lines 312-315 suggest KEGG and Wikipathways are generally interchangeable if the same signaling subnetworks are present. However, differences in relevance ranking between KEGG and Wikipathways are observed for the same biological pathway in Fig 2A. What explains the difference between KEGG and Wikipathway relevance scoring? It might be helpful to comment on this in the text.

- For example, relevance scores for SP-600125 in Figure 2A shows WP-insulin signaling as significant, while KEGG-insulin signaling as insignificant.
 - Similarly, WP: PI3K-AKT-mTOR signaling shows high relevance (~ 3-4), while the similar KEGG: mTOR signaling pathway ranks with 0 relevance.
5. Lines 123-124: The paper lacks detail on the variables underlying the relevance scoring model imported from g:profiler. Background information on the g:profiler scoring model should be included in the paper to highlight its gene-centric nature and help users investigate sources of inflated, missing, or conflicting relevance ranking during manual investigation. Include abbreviated highlights of the scoring algorithm as implemented in PTMNavigator. Indicate whether this is a gene-centric or gene-centric-redundant implementation. At minimum, specify if pathway matching is performed at the gene level and whether PTMs are accounted for in relevance scoring.
6. Line 192 - 193: indicate the intended target of Cobimetinib in the main text to facilitate comparison to Sorafenib along the RAF-MEK-ERK pathway.
7. Line 156: I suggest elaborating on specific criteria regarding location of phosphosite regulations mapped onto BDNF signaling that appeared false. Such details should provide a framework to more novice users of the tool during manual evaluation of results as suggested in the discussion section. The current description for determination of a false match was too vague.
8. Lines 329-330; Figure 3, S4, S6, and most all other pathway figures: +/- p for phosphorylation or dephosphorylation derived from KEGG Pathways needs to be noted in legends and somewhere in text as they are prevalent in PTMNavigator output. Near line 329-330, explicit distinction between the +/-p edge annotations from KEGG and the experimental PTM data mapped to nodes should be explicitly stated to inform users and readers of their differences. Additional comment on the meaning and shortcomings of the +/-p edge annotations near line 329-330 may improve discussion on the limitation of KEGG pathway diagrams.
9. Figure S6- S15: Numbers inside regulated PTM sites should be defined in legend or caption. I assume these numbers indicate counts of distinct regulated phosphosites for a given protein, but this should be explicit.
10. Figure S3: Figure legend is confusing with plot.
 - No unmapped gene identifiers appear on the plot. However, the legend indicates unmapped gene identifiers will be represented with a transparent gray line or fill.
 - The legend also states 3,112 Uniprot accession numbers are unmapped. Are these represented in the plot?
 - I suggest clarifying distinction between mapped and unmapped entries, such as by moving the 'mapped' color swatch to the left column in the legend, if correct.
11. Figure 5 and S5: What is the explanation for the multiple MYPT1_S507 and GRLF_S589 drug potency curves? Explanation of multiple curve plotting for the same site should be mentioned explicitly in any of the main text, figure legend, or materials and methods.
12. Supp Fig. S6 - S12, S13-S14: Rationale for only displaying regions enclosed in dotted lines should be explicitly stated in text (either figure caption, main text or methods). Justification of the cropped zoom can help less experienced users understand how to display PTM subnetworks.
13. Figure 2: missing figure caption.
14. Lines 152-154 and Fig 2b: The plot in Fig 2B shows the fraction of a pathway matched between experimental and reference data. The plot clearly helps distinguish between BDNF signaling and PI3K-AKT-mTOR signaling on the basis of node matches where relevance scores show an inverse relationship. As such, analysis of 'fraction of pathway matched' such as Fig 2B may improve interpretation of g:profiler relevance scoring. Analyzing this possibility will strengthen the manuscript. Lines 152 - 154 indicate that this plot was generated from PTM Navigator. However, I could not find or may have missed this plot. Please append text to indicate how the plot was generated or link to custom code. If the plot is not already a feature of PTM Navigator, its addition is highly suggested.
15. Lines 74-80; 322-323: (continuation of what is mentioned in major comments) The manuscript states that PTM Navigator can be updated when better ranking procedures are developed (lines 322-323). Two alternative ranking models already exist as outlined in Krug et al. MCP, 2019. While the alternative scoring was introduced in lines 74-80, claims of superiority of g:profiler for PTM pathway analysis in lines 322-333 should be modified. If the scoring model will not be changed, at a minimum the paper should be revised to indicate g:profiler was used for its ease of implementation despite inferior performance relative to PTM-signature scoring or gene-centric-redundant scoring (Krug et al. MCP, 2019).

Reviewer #3

(Remarks to the Author)

Reviewer #4

(Remarks to the Author)

Summary

The authors describe in this manuscript a tool for visualizing phosphoproteomics data within curated pathway diagrams from KEGG and WikiPathways. The tool can visualize qualitative or quantitative phosphosite changes, and integrates relevant information from PhosphositePlus and other sources to facilitate the interpretation of these changes. The authors demonstrate the utility of this tool through several case studies: 1) it can facilitate manually discarding enriched pathways that do not seem biologically plausible; 2) it helps reveal the flow of signal through pathways and differences in signal flow between perturbations; 3) it expedites the process of proposing new links in pathways based on associations in omics data. While this is a useful resource for the signaling and proteomics communities, the tool and the accompanying manuscript are somewhat limited in their scope and may be more suitable for a more specialized audience.

Major concerns

While I believe it is useful to have a way to draw pathways with quantitative information from phosphoproteomics studies, the major concern I have is that, as of yet, the tool does not facilitate the derivation of novel information. Furthermore, the advance made either in terms of method/tool development or biological discoveries remains limited. For the tool to comprehensively facilitate pathway-level analysis of phosphoproteomics data, it would benefit from incorporating additional information and common analyses. For the benefit of the authors I include below some example analysis that would improve the usefulness of a such a tool:

- 1) Performing differential expression analysis. Currently the user must do this themselves, which slows down the overall process.
- 2) Incorporating kinase-substrate predictions from tools such as NetworKIN and the Kinase Library (10.1093/nar/gkm902, 10.1038/s41586-022-05575-3). Since the majority of differentially regulated phosphosites in a given phosphoproteomics experiment will have no annotated upstream kinase, these predictions can greatly facilitate the interpretation of results. This would be especially true when results are visualized within a pathway diagram, where predicted kinases can be viewed within their pathway context.
- 3) Incorporating estimated changes in kinase activity using tools such as KSEA, KEA3, etc. (10.1093/bioinformatics/btx415, 10.1093/nar/gkab359). In the current implementation of the tool, users can only infer pathway signal flow by tracking changes in individual phosphosites. Given kinases are the major conduits for signal flow, estimates of their regulation would be useful.

The pathway diagrams contain directed edges from kinases to other proteins, indicating that the kinase phosphorylates a subset of the protein's phosphosites. The authors have provided access to this information through a tooltip, which summarizes PhosphositePlus annotation including upstream kinases. However, it would be more informative if specific kinase-phosphosite relationships were incorporated as edges within the diagram itself. This would allow the user to immediately track signal flow from kinases to their substrates; currently they must manually click on each substrate phosphosite to do this.

In the introduction the authors highlight several limitations of pathway enrichment analysis, such as that it does not distinguish between randomly distributed changes vs discernible signal flow, and that it considers all genes as equally important to pathway activity. While the tool helps tackle these problems, it does not overcome them in a systematic manner; the burden is still on the researcher to manually inspect and judge enriched pathways.

Minor points

It would be helpful if phosphosites were automatically labeled with their residue and position (e.g. Y187 on MAPK1). Currently the user has to manually click on phosphosites to see this information.

The authors suggest that PTMNavigator can be used to uncover unknown relationships between PTMs and pathway components, and hence extend existing pathways (e.g. lines 24-26 in the abstract). The unknown relationships that they propose (e.g. an unknown interaction between MAP2K1/2 and GRLF1) are based on correlations in phosphoproteomics data. Without additional evidence for these relationships (for example in vitro kinase assays showing MAP2K1/2 can phosphorylate GRLF1) I would be hesitant to extend a pathway diagram in such a way. I recommend that the authors weaken the language used, for example by saying that PTMNavigator "can aid in proposing putative new links in pathways".

Summary

The authors describe in this manuscript a tool for visualizing phosphoproteomics data within curated pathway diagrams from KEGG and WikiPathways. The tool can visualize qualitative or quantitative phosphosite changes, and integrates relevant information from PhosphositePlus and other sources to facilitate the interpretation of these changes. The authors demonstrate the utility of this tool through several case studies: 1) it can facilitate manually discarding enriched pathways that do not seem biologically plausible; 2) it helps reveal the flow of signal through pathways and differences in signal flow between perturbations; 3) it expedites the process of proposing new links in pathways based on associations in omics data. While this is a useful resource for the signaling and proteomics communities, the tool and the accompanying manuscript are somewhat limited in their scope and may be more suitable for a more specialized audience.

Major concerns

While I believe it is useful to have a way to draw pathways with quantitative information from phosphoproteomics studies, the major concern I have is that, as of yet, the tool does not facilitate the derivation of novel information. Furthermore, the advance made either in terms of method/tool development or biological discoveries remains limited. For the tool to comprehensively facilitate pathway-level analysis of phosphoproteomics data, it would benefit from incorporating additional information and common analyses. For the benefit of the authors I include below some example analysis that would improve the usefulness of a such a tool:

- 1) Performing differential expression analysis. Currently the user must do this themselves, which slows down the overall process.
- 2) Incorporating kinase-substrate predictions from tools such as NetworkKIN and the Kinase Library (10.1093/nar/gkm902, 10.1038/s41586-022-05575-3). Since the majority of differentially regulated phosphosites in a given phosphoproteomics experiment will have no annotated upstream kinase, these predictions can greatly facilitate the interpretation of results. This would be especially true when results are visualized within a pathway diagram, where predicted kinases can be viewed within their pathway context.
- 3) Incorporating estimated changes in kinase activity using tools such as KSEA, KEA3, etc. (10.1093/bioinformatics/btx415, 10.1093/nar/gkab359). In the current implementation of the tool, users can only infer pathway signal flow by tracking changes in individual phosphosites. Given kinases are the major conduits for signal flow, estimates of their regulation would be useful.

The pathway diagrams contain directed edges from kinases to other proteins, indicating that the kinase phosphorylates a subset of the protein's phosphosites. The authors have provided access to this information through a tooltip, which summarizes PhosphositePlus annotation including upstream kinases. However, it would be more informative if specific kinase-phosphosite relationships were incorporated as edges within the diagram itself. This would allow the user to immediately track signal flow from kinases to their substrates; currently they must manually click on each substrate phosphosite to do this.

In the introduction the authors highlight several limitations of pathway enrichment analysis, such as that it does not distinguish between randomly distributed changes vs discernible signal flow, and that it considers all genes as equally important to pathway activity. While the tool helps tackle these problems, it does not overcome them in a systematic manner; the burden is still on the researcher to manually inspect and judge enriched pathways.

Minor points

It would be helpful if phosphosites were automatically labeled with their residue and position (e.g. Y187 on MAPK1). Currently the user has to manually click on phosphosites to see this information.

The authors suggest that PTMNavigator can be used to uncover unknown relationships between PTMs and pathway components, and hence extend existing pathways (e.g. lines 24-26 in the abstract). The unknown relationships that they propose (e.g. an unknown interaction between MAP2K1/2 and GRLF1) are based on correlations in phosphoproteomics data. Without additional evidence for these relationships (for example in vitro kinase assays showing MAP2K1/2 can phosphorylate GRLF1) I would be hesitant to extend a pathway diagram in such a way. I recommend that the authors weaken the language used, for example by saying that PTMNavigator "can aid in proposing putative new links in pathways".

Version 1:

Reviewer comments:

Reviewer #1

(Remarks to the Author)

The authors have addressed many of the points raised during the first round of revision, this has strengthened PTMNavigator as a tool for the community. After engaging with the website I have several remaining requests, comments and concerns.

1. When loading my own phospho dataset, I followed the format suggestion provided in 'examples'-'fold_change_data'-'psite_level_data.csv'. I was unable to see any enrichment scores for pathways, no PhosphoSitePlus custom pathway generation, nor any scoring with other analysis tools in my uploaded data. If this is a result of formatting issues, or insufficient input data, this is unclear to me as a user. If the analysis was not yet complete, the status and progress of the task was unclear. Given my struggle to use the tool successfully, I suggest more clarity is provided to the user to abide by necessary formatting. More troubleshooting direction would be helpful beyond the provided text "Did you just upload this dataset? If it still is not here after a few minutes, something probably went wrong. Please contact a site administrator."

2. I could not successfully load any datasets from the ProteomicsDB_Data list. This was true across different browsers and for different projects listed.

3. It is helpful to see analysis tools like PTM SEA integrated into the browser, however there is no associated metadata available for the user to see details on how the enrichment analysis was performed. For example, in PTM SEA there are

several parameters the user can adjust including normalization method, overlap minimums, weighting, etc. PTMNavigator should provide information equivalent to a parameters output file for the enrichments that are performed on the datasets. This is helpful for FAIR data purposes. Furthermore, can the list of protein/geneIDs that are actually being used in the enrichment be added to the data tables? This way users can see directly which protein nodes are contributing to the enrichment score. They can then use these nodes as filtering criteria for their pathway searches using the provided 'Filter pathways by Protein List' feature.

4. The Expand function in the tool would be more useful if there were additional levels of selectivity beyond 'expand all'. For example, can the user only expand regulated sites, and/or sites for only the selected node(s)? Subsetting the data before expansion is a useful feature, especially for nodes with many PTM sites.

5. It is not clear how multiple studies are integrated into the pathway maps or used to generate the pathway scores. Are multiple datasets or experiments considered when generating the pathway enrichment scores? If yes, how is this done? How can users readily see which sites are being mapped from which experiment? There should be more features added to understand when the user is viewing combined data or data from a single experiment, and how data integration is occurring.

6. At present it is difficult to see all of the phosphosites associated with a protein node. If the user clicks on the node, no PTM data is listed in the 'currently selected tab.' Is it possible to add a table view to the current selection where all of the associated phosphosites for a selected protein node are listed? This would also be a place to directly see, in table form, the PTM regulation occurring from different experiments that are loaded into the pathway.

7. The figures displayed in Figure 3B/C/D were generated with other features of ProteomicsDB, correct? If so, the authors should indicate in the manuscript which ProteomicsDB analysis features were used. If the data analysis is outside the scope of ProteomicsDB, then the manuscript should state that the analysis was done independent of the webtool.

(Remarks on code availability)

Reviewer #2

(Remarks to the Author)

The revised manuscript incorporates major changes asked by reviewers. We appreciate the work invested to incorporate kinase activity inference and pathway scoring algorithms, as well as improve flexibility of the visualizations. We are satisfied with the revised version and suggest proceeding with publication.

That said, we noticed a few minor technical issues with usability of the new features, listed below:

1. Contrary to the provided instructions, CTRL click seems to deselect nodes from the view. If this is supposed to be the functionality, please make the instructions clearer.
2. Kinase relationships shown by red arrows only appear if the user selects 'Expand All' in the right-click menu in addition to 'Show kinase and substrate relationships'. Please add this to the instructions on the site or fix this default.
3. Is it possible to avoid having the figure disappear when transitioning between viewing and editing modes? It is somewhat inconvenient to have it disappear when switching modes.

We recommend these three issues are fixed prior to publication.

(Remarks on code availability)

We reviewed the code in the first submission, but not now.

We tested the new tool.

Reviewer #3

(Remarks to the Author)

(Remarks on code availability)

I reviewed the code in the first version of the manuscript. I checked the output of the new features worked as expected. There is a readme in the features code.

Reviewer #4

(Remarks to the Author)

I appreciate the work of the authors in revising the manuscript. They addressed the majority of my suggestions and have substantially enhanced the functionality and usability of the tool, making it a better resource for the research community. Some notable improvements include the addition of edges of kinase-substrate interactions, phosphosite labels, and the premise of automatically performing pathway enrichment and ranking pathways using ssGSEA.

While testing the tool I did encounter some limitations with the new features. First, the ssGSEA enrichment seems to be buggy or errorprone. For example, I loaded one of the supplied datasets (Session ID: 0123456789ABCDEF0123456789ABCDEF, uploaded datasets: BekkerJensen2020_AKTi_AKT-1-2, Experiment Filter:

BekkerJensen2020_AKTi_AKT-1-2) and selected the most highly ranked pathway (3q29 copy number variation syndrome (WP4906) [Score=4.38]). However, on inspection of the pathway I could not find any differentially regulated phosphosites, which seems in direct contradiction with it being the most highly ranked pathway. Second, while it is useful to display kinase activity estimates on the pathway diagram, these were visually not very salient; furthermore, it was unclear to me what threshold had been used to select active kinases. This could be displayed either in the tool, or the user could be allowed to select the threshold themselves. Given that multiple kinase enrichment tools have now been provided in PTMNavigator, it would also make sense to allow the user to select the specific source of kinase enrichment. Importantly, the tool still does not address a critical issue in pathway enrichment analysis: how to systematically prioritize pathways with discernable signal flow, rather than those with randomly distributed changes. The tool only provides means for a user to attempt this by manual inspection.

Overall, while the authors have substantially improved their tool and the accompanying manuscript, I believe this represents a modest methodological and biological advance.

(Remarks on code availability)

Version 2:

Reviewer comments:

Reviewer #1

(Remarks to the Author)

I am very happy to read that the authors considered all the suggestions and updated the website accordingly. These edits, once successfully implemented, will help make PTM navigator a more clear useful tool. I am eager to see this webtool operate to its full potential as this is a tool I would recommend to my lab and collaborators.

That being said, I was unable to experience some of the improvements mentioned in the rebuttal. For example, no where could I find any enrichment for any dataset, not my own data or the preloaded dataset files. Previously I could see scoring in the pathways and in the Enrichment Analysis with the provided datasets, I just couldn't get it to work with my own data. Now I see no enrichment anywhere with any dataset. The only thing that happened during my interaction was a list of all possible pathways was loaded, with no scoring. I could not find the Figure 4 revision with threshold scoring anywhere on the site. If perhaps this is my own user error, I spend 30 min attempting to figure it out with no success, so the intuitive user experience may need to be addressed. Without any scoring, the entire utility of the tool is decreases as now I have 1000+ unranked pathways to sort from to visualise my data. I therefore would not recommend or use the tool in its present state.

An additional suggestion, I find myself frequently lost in scrolling to try and find the pathway that has floated away in the zoom space. Is there some way to recover the original zoom and x y z placement of the image with a right click?

(Remarks on code availability)

Reviewer #4

(Remarks to the Author)

The reviewers have addressed the previous concerns

(Remarks on code availability)

Point to point response to reviewer comments

Reviewer #1 (Remarks to the Author):

Manuscript review points

This manuscript describes PTMNavigator which as a new user-friendly data analysis tool to visualize quantitative phosphorylation data on signaling pathway maps. The article is clear and well written. The PTMNavigator tool will certainly be helpful to many users. However, it is not clear to me that this manuscript presents a significant enough advancement in the field to warrant publication in this journal. The manuscript at present is better suited for another journal. There are many desired features that would need to be added to the tool to make this resource a significant advancement, many of these features are addressed by the authors themselves in the discussion section. It is clear in the manuscript that extensive manual interpretation is required to draw conclusions from the pathway maps. In many cases this data could already be scored and quantified by the tool in order to help users reach biological interpretations faster and easier. Some specific comments:

We thank the reviewer for the overall positive feedback and constructive comments. Before we begin our point by point response, we would like to outline the most notable additions we made to our software: i) Pathway Editor: Users can now create new or modify existing pathway diagrams by adding and removing nodes and edges, as well as rearranging them on the canvas. ii) Enrichment Server: We implemented a service application that hosts a number of pathway and kinase enrichment analysis algorithms for phosphoproteomics data such as ssGSEA, KSEA, ROKAI, PHONEMeS, Kinase library and KEA3. Using this service, we now provide new options for sorting pathways by putative relevance such as Gene-Centric Redundant ssGSEA Scores. iii) Data-driven putative kinase signaling pathways: Based on the output of kinase enrichment algorithms, we generate a data-driven putative kinase signaling pathway using the PHONEMeS tool. This tool uses a prior knowledge network of protein-protein interaction and kinase-substrate relationships to reconstruct a signaling network, which can subsequently be viewed and edited in PTMNavigator. iv) Context Menu for customization of visualizations: We added a context-menu ('right-click menu'), which incorporates several new optional visualization features such as highlighting enriched kinases, labeling PTM Nodes with their amino acid residue and position and displaying kinase-substrate relationships from PhosphoSitePlus as additional edges.

1. The authors point out the challenge of interpreting pathway enrichments that are not biologically relevant for the samples analysed. These pathways are enriched based on a small subset of proteins (or subnetwork) that is actually a shared feature with other, more relevant pathways. Why not provided weighted protein-centric scoring to identify these subnetworks or protein subsets that are shared across multiple pathways? An algorithm can surely identify and visualize these overlaps better than manual curation of the different pathway hits. Automating the identification of such shared subnetworks will identify protein drivers embedded within larger pathways and will help avoid misleading biological insight from irrelevant pathway hits.

We agree that identifying shared subnetworks is a desirable feature. Unfortunately, initial experiments with such algorithms did not produce useful results, likely due to the inconsistency between protein-protein links in pathways even within the same resource. Nevertheless, the integration of kinase enrichment algorithms combined with PHONEMeS into our analysis pipeline addresses the need for identifying protein drivers.

2. Pathway expansions performed in Fig 5D is helpful and should already be a feature in this tool. If

it is not possible to expand the visual, at least the metadata associated with the sites should be added to the existing pathway maps.

This was indeed a frequently requested feature from our users. As outlined in our revision summary above, creating and editing existing pathways is now possible in PTMNavigator. We also added the option to visualize additional metadata for phosphosites via the new context menu.

3. Can the extent signaling crosstalk between pathways be quantified?

We do not explicitly quantify signaling crosstalk and consider an automated method of doing so outside the scope of the current manuscript. Nevertheless, PTMNavigator can assist in addressing such questions by way of the different pathway enrichment analyses as well as the data-driven pathways from PHONEMeS which then require manual evaluation.

4. Kinase activity inference should be a feature in this tool. It is a helpful readout which can already been found in tools like PTMgsea. Connecting evidence of kinase activation with different pathway signaling nodes in a visualisation would be a unique contribution from this tool.

We fully agree that kinase activity inference improves the interpretation of signaling pathways. We have now integrated multiple kinase activity inference algorithms (KSEA, KEA3, KSTAR) into our tool and also added a visualization of the results within the pathway diagram together with the kinase-substrate relations themselves.

5. Phosphorylation site values go up or down, but what about protein activity? Are these increase or decreases in phosphorylation associated with activation or deactivation of protein function? Certainly this info is only known for a small subset of proteins, but it should be accessible and annotated for those sites. Ultimately this would be a biological interpretation that is sought after in quantitative phosphoproteomics data.

This functionality was already available in PTMNavigator but was perhaps not sufficiently highlighted. If available, functional annotations from PhosphoSitePlus, including effects on protein function and activity, are available in the tooltip of the PTM nodes. We have clarified this in the text as follows (Lines 134-138):

*“Where available, additional information on PTM sites is displayed as part of a tooltip and kinase-substrate relationships can be visualized in the form of additional arrows in the diagram. To this end, we imported a list of functional annotations and upstream regulators from PhosphoSitePlus and a list of predicted phosphosite functional scores calculated by Ochoa *et al.*³³ into ProteomicsDB.”*

6. Integration of a feature that allows the magnitude of phosphorylation changes to be highlighted over proteome changes for any given protein would be a very interesting. You could visualize which sites are trending with protein change and which have dynamic regulation beyond protein level changes.

The simultaneous visualization of PTM and Protein-Level data was actually already possible in the first version of PTMNavigator, however, we did not highlight in our manuscript that this feature

exists. We have now added a supplemental figure (Fig. S5) which exemplifies this and added the following text to the manuscript (Lines 138-140):

“It is also possible to visualize protein-level data in combination with PTM-level data (see Suppl. Fig. S5 for an example, where this has been exemplified using a dataset from *Lee et al.*³⁴).”

7. There are already other tool that can visualize phosphorylation changes within protein networks, such as PhosphoPath within Cytoscape.

We are aware of other PTM visualization tools as highlighted in the introduction. Still, PTMNavigator comes with several features that complement and surpass the functionalities of PhosphoPath and other such tools. First, users do not need to install any software to use our tool. For example, in the case of PhosphoPath, we found it challenging to find a version of Cytoscape where the PhosphoPath plugin works. Second, PTMNavigator makes use of the curated pathway layouts from the original resource, which we think are superior to the layouts generated by Cytoscape. Last, with the new functionalities highlighted in the revision summary above, PTMNavigator becomes a powerful data analysis tool with functionalities, to the best of our knowledge, not available in any other tool. Together, PTMNavigator is currently the most user-friendly and convenient way for evaluating PTM signaling as well as sharing datasets between PCs and people.

Reviewer #2 (Remarks to the Author):

Summary:

PTMNavigator offers a novel and user friendly interface to map individual post-translational modifications with fold change or drug potency values onto proteins along pathway diagrams imported from KEGG and wikipathways with links to functional annotations from PhosphositePlus and from Ochoa et al (2020). The tool includes the most pathway diagrams of any tool to date and best visual presentation of sites along a pathway when compared to Cytoscape (PTMOracle, Phosphopath), Phosphomatics, KEGGViewer, Wikipathways and PhosphositePlus. The paper successfully demonstrated the visualization capabilities of the tool with internal and external datasets. The added benefit to the field is exemplified in Supplementary Figure 4, which nicely illustrates how signaling differs between cell lines following dasatinib treatment. Issues with the relevance scoring were discovered during analysis of the test data and were only partially acknowledged. Relevance scoring issues should be explicitly addressed in the revision as suggested below to mitigate overreliance on the ranking functionality and acknowledge previous publications outlining issues of gene-centric ranking models for PTM analysis. As it stands, most users who are not signaling experts and study mouse or human phosphorylation pathways will benefit from starting with alternative tool PTM-SEA (Krug et al MCP, 2019) for pathway identification, followed by visualization in PTMNavigator. However, domain experts and users studying the eight other organisms uniquely represented in PTMNavigator will benefit from PTMNavigator visualization in concert with literature searching for pathway identification. Overall, this work adds a useful visualization tool to the fields of PTM signaling and systems biology.

We thank the reviewer for the kind remarks. We agree that it is beneficial to use PTM-SEA and PTMNavigator in combination. PTM-SEA analysis is now integrated into our workflow, at least for Homo sapiens datasets, eliminating the need to use two separate tools. Before we begin our point by point response, we would like to outline the most notable additions we made to our software: i)

Pathway Editor: Users can now create new or modify existing pathway diagrams by adding and removing nodes and edges, as well as rearranging them on the canvas. ii) Enrichment Server: We implemented a service application that hosts a number of pathway and kinase enrichment analysis algorithms for phosphoproteomics data such as ssGSEA, KSEA, ROKAI, PHONEMeS, Kinase library and KEA3. Using this service, we now provide new options for sorting pathways by putative relevance such as Gene-Centric Redundant ssGSEA Scores. iii) Data-driven putative kinase signaling pathways: Based on the output of kinase enrichment algorithms, we generate a data-driven putative kinase signaling pathway using the PHONEMeS tool. This tool uses a prior knowledge network of protein-protein interaction and kinase-substrate relationships to reconstruct a signaling network, which can subsequently be viewed and edited in PTMNavigator. iv) Context Menu for customization of visualizations: We added a context-menu ('right-click menu'), which incorporates several new optional visualization features such as highlighting enriched kinases, labeling PTM Nodes with their amino acid residue and position and displaying kinase-substrate relationships from PhosphoSitePlus as additional edges.

Major comments:

A critical issue with the paper is the partial presentation of shortcomings of the gene-centric relevance ranking model imported from g:profiler and missing acknowledgement of Krug et al MCP 2019 for previous presentation of ranking issues and identification of two better ranking algorithms.

Our learnings about the shortcomings of gene-centric rankings (Lines 64 – 74 in the original manuscript) indeed stem predominantly from Krug et al., and we mentioned and referenced their method in the subsequent part of our manuscript (Lines 74 – 77). However, we did not make it explicit that Krug et al. are also the resource for the former statements, which we now remedied by an additional citation. Moreover, we have now included both the gene-centric (redundant) and PTM-centric ssGSEA analysis into our tool and have abandoned g:Profiler analysis.

We have updated the text as follows (Lines 61-73):

“To place all regulated peptides into the context of signaling networks, a commonly employed method is pathway enrichment analysis (PEA)¹³⁻¹⁵, implemented in tools such as g:Profiler¹⁶. While very useful for the analysis of proteome expression changes, this approach has several shortcomings in the context of PTM analysis, as has been pointed out by Krug et al.¹⁷. The authors introduced two algorithms that address the problems of traditional PEA: One is gene-centric redundant (GCR) single-sample gene set enrichment analysis (ssGSEA), in which genes are potentially counted multiple times during the calculation of enrichment scores, depending on how many regulated PTMs they harbor. The second is PTM-signature enrichment analysis (PTM-SEA), which uses a specifically assembled database of associations between sites of PTMs and perturbations, pathways, or kinase activities (PTM signatures database; PTMsigDB) and thus can calculate enrichments on PTM level. This constitutes a valuable complementary approach to conventional PEA and both methods have shown superior performance compared to non-redundant gene-centric PEA.”

First, the paper suggests manual inspection of middle to high ranking pathways will contain the truly perturbed pathway amidst PEA artifacts (Lines 96 - 99; 317-320). However, this is not always the case, as exemplified in Temsirolimus, which targets mTOR late in the pathway and shows almost 0 relevance (Lines 174 - 189 and Fig 2A). The paper should explicitly state this issue to point readers toward inspection of all pathways despite relevance ranking.

We have addressed this in Lines 356-358 of the revised manuscript:

“On the other hand, also pathways with non-significant scores are often worth examining and additional resources, such as previous literature, should be considered when selecting diagrams for visualization.”

Second, the paper suggests the KEGG and wikipathways are interchangeable, however the relevance rankings contradict this claim. The paper should explain to readers why this may be the case.

We apologize for the unclear wording. What we were trying to say is the following: “If KEGG and WikiPathways have a pathway with the same/similar name, sometimes KEGG will represent your dataset better, sometimes WikiPathways. There is not really a way to tell in advance which one it will be (the relevance ranking is in our opinion suggestive, but not conclusive), so we advise to look at both”.

We have rephrased the respective paragraph (Lines 335-349 in the revised manuscript).

“There are usually multiple pathways that depict a signaling cascade of interest, and choosing a single one can seem arbitrary. We’d like to note that for the purpose of conciseness, in this manuscript we always selected a single KEGG or WikiPathways diagram for visualization of a dataset, while in practice we recommend that the user should look at all diagrams of highly enriched pathways to get an idea of how interchangeable the visualizations are and see if specific subnetworks are shared between them. For example, the visualization of MEK inhibition, for which we chose ‘Regulation of actin cytoskeleton – WP51’ (Fig. 6A), would for our purpose have worked just as well with the eponymous KEGG diagram ‘hsa04810 – Regulation of actin cytoskeleton’, as it contains the same subnetwork of SOS1, RAF1, MAPK1/3, GRLF1, and MYPT1. Meanwhile, for other datasets or visualization purposes one diagram might be preferable over the other, and it is usually not possible to decide that without inspecting both. This is also the case if one of the pathways has a substantially higher relevance score – this can simply happen because the other pathway contains more additional proteins that ‘water down’ the score, however, that pathway can still be the better choice for visualization if the subnetwork of interest is more appropriately represented.”

Third, Krug et al MCP (2019) outlined inferior performance of gene-centric approaches for pathway scoring relative to either gene-centric-redundant or PTM signature based scoring. Despite previously published knowledge, PTMNavigator implemented the inferior gene-centric scoring with g:profiler and arrived at similar conclusions without citing the previous findings. Further, in lines 72-73, the paper indicated aggregation of PTMs to protein level modifies pathway enrichment analysis for PTM signaling, but the variable functionality of PTMs questions the validity of this approach. Please add citations supporting the questionable impact of such an approach. Otherwise, such a claim distorts prior literature showing superior performance of gene-set-redundant scoring relative to gene-centric scoring (Krug et al. MCP (2019)).

We wholeheartedly agree with the reviewer that the method by Krug et al. is superior and we were, from the start, very much in favor of it. In our first submission, we used g:Profiler out of convenience because it is available as an API, and by using this option we avoided having to implement and host a method ourselves. For our revision, we made the effort to implement an enrichment service, which now enables interfacing PTMNavigator with methods such as the gene-centric-redundant and PTM-SEA scoring algorithms. Furthermore, we rewrote the chapter that was focusing on g:Profiler enrichment (formerly ‘Interpreting Pathway Enrichment Analysis Results

with PTMNavigator', *now entitled* 'Interpreting Pathway Enrichment Analysis and Pathway Visualization Results with Synergize in PTMNavigator') *and redesigned the accompanying Figure (which is now Figure 3).*

Fourth, the paper states in lines 322-323 that PTMNavigator will be updated with better scoring models once available and in line 317 that PEAs provide currently the best starting point for pathway relevance sorting. Better scoring models are available and should be incorporated, or otherwise modify the statement. Either the gene-centric-redundant or PTM-signature-set scoring outperform gene-centric scoring per Krug et al MCP (2020). The tool will be significantly improved if the PTM-signature sets from PTM-SEA along with DeCryptM Kinase inhibitor signatures are imported into PTMNavigator and a PTM-signature score implemented. Given these implementations may require significant effort, an alternative option is to modify g:profiler implementation to test a gene-centric-redundant model as tested in Krug et al. MCP (2020); or modify statements in lines 317, 322-323.

As stated above, we now perform the ranking of pathways using gene-centric redundant scoring. We would like to point out that PTM-SEA cannot currently be used for this purpose, as PTMSigDB only contains 14 WikiPathways and no KEGG signatures. Still, PTM-centric signature enrichment can provide valuable insights for PTMNavigator users. Therefore, we support it in the new version. DecryptM kinase inhibitor signatures in the same style as PTMSigDB signatures would indeed be another useful addition. We do not have these signatures yet, but members of our research lab are actively working on creating a large number of such signatures and we will integrate them once available.

In our view, whilst some methods might be more useful than others, all of these methods are 'far from ideal' and the results must 'be taken with considerable caution' (as stated in Lines 316-317 in the original manuscript). This is due to reasons such as overlapping and inconsistent pathway definitions, context-specificity (pathways can work differently in different cell types or cell states), or simply limited knowledge of the underlying molecular biology. In our opinion, it happens too often that a pathway or signature enrichment is performed as the last step in the analysis of a dataset and then simply plotted and published as is, in order to connect some sort of 'biological meaning' to the data. Instead, such an analysis should rather serve as a starting point for manual in-depth investigation or new experimentation. While PTMNavigator also suffers from the drawbacks mentioned above (as we elaborate on in the Discussion section), we believe that it can assist in these subsequent steps.

Minor comments:

1. line 89 claims "quick" examination of pathway engagement, which is not necessarily the case if extensive manual investigation is still required. I suggest changing "quickly" to "visually."

We thank the reviewer for this suggestion and have adapted the suggested change.

2. Lines 96 - 99: Change to "allow users to visually inspect both high and low ranked pathways." Lines 174 - 189 and Fig 2A indicate Temsirolimus targets mTOR pathway however the mTOR pathway receives a 0 or near insignificant relevance score. Therefore, users should be aware that even the lowest ranking pathways may represent biological truth.

We thank the reviewer for this suggestion and have adapted the suggested change.

3. Lines 318 - 320: The paper notes that initial relevance rankings help users prioritize pathway engagement, but that some high-ranking pathways are not biologically relevant and are instead a PEA artifact. This statement should be extended to additionally indicate some 'low-ranking or unranked pathways may instead be of true biological relevance' in light of the observations above.

We agree with the reviewer that this should be pointed out more clearly. We have updated the text as follows (Lines 356-358 in the revised manuscript):

“On the other hand, also pathways with non-significant scores are often worth examining and additional resources, such as previous literature, should be considered when selecting diagrams for visualization.”

4. Figure 2A (lines 624-625) and lines 312 to 315 contradict each other regarding the similarity between KEGG and Wikipathways. Lines 312-315 suggest KEGG and Wikipathways are generally interchangeable if the same signaling subnetworks are present. However, differences in relevance ranking between KEGG and Wikipathways are observed for the same biological pathway in Fig 2A. What explains the difference between KEGG and Wikipathway relevance scoring? It might be helpful to comment on this in the text.

- For example, relevance scores for SP-600125 in Figure 2A shows WP-insulin signaling as significant, while KEGG-insulin signaling as insignificant.

- Similarly, WP: PI3K-AKT-mTOR signaling shows high relevance (~ 3-4), while the similar KEGG: mTOR signaling pathway ranks with 0 relevance.

We thank the reviewer for pointing this out. We did not intend to state that KEGG and WikiPathways are generally interchangeable and apologize for the misunderstanding. In order to convey this message better, we have revised our manuscript (Lines 340-349) as follows (changes in bold):

“For example, the visualization of MEK inhibition, for which we chose ‘WP51 – Regulation of actin cytoskeleton’ (Fig. 5A), would **for our purpose** have worked just as well with the eponymous KEGG diagram ‘hsa04810 – Regulation of actin cytoskeleton’, as it contains the same subnetwork of SOS1, RAF1, MAPK1/3, GRLF1, and MYPT1. **Meanwhile, for other datasets or visualization purposes one diagram might be preferable over the other, and it is usually not possible to decide that without inspecting both. This is also the case if one of the pathways has a substantially higher relevance score – this can simply happen because the other pathway contains more additional proteins that ‘water down’ the score, however, that pathway can still be the better choice for visualization if the subnetwork of interest is more appropriately represented.**”

5. Lines 123-124: The paper lacks detail on the variables underlying the relevance scoring model imported from g:profiler. Background information on the g:profiler scoring model should be included in the paper to highlight its gene-centric nature and help users investigate sources of inflated, missing, or conflicting relevance ranking during manual investigation.

Include abbreviated highlights of the scoring algorithm as implemented in PTMNavigator.

Indicate whether this is a gene-centric or gene-centric-redundant implementation.

At minimum, specify if pathway matching is performed at the gene level and whether PTMs are accounted for in relevance scoring.

We do not use g:Profiler anymore but added more detailed information about the enrichment algorithms that we use now. See the Methods section of the revised manuscript (subsection ‘Implementation of a Service for Enrichment Analyses’).

6. Line 192 - 193: indicate the intended target of Cobimetinib in the main text to facilitate comparison to Sorafenib along the RAF-MEK-ERK pathway.

We thank the reviewer for pointing this out – we forgot to mention this in the main text. We have now added also in the text that Cobimetinib inhibits MEK (Line 219).

7. Line 156: I suggest elaborating on specific criteria regarding location of phosphosite regulations mapped onto BDNF signaling that appeared false. Such details should provide a framework to more novice users of the tool during manual evaluation of results as suggested in the discussion section. The current description for determination of a false match was too vague.

We agree with the reviewer and thank them for pointing this out. Since this section has undergone major revision, we did not address this specific comment.

8. Lines 329-330; Figure 3, S4, S6, and most all other pathway figures: +/- p for phosphorylation or dephosphorylation derived from KEGG Pathways needs to be noted in legends and somewhere in text as they are prevalent in PTMNavigator output. Near line 329-330, explicit distinction between the +/-p edge annotations from KEGG and the experimental PTM data mapped to nodes should be explicitly stated to inform users and readers of their differences. Additional comment on the meaning and shortcomings of the +/-p edge annotations near line 329-330 may improve discussion on the limitation of KEGG pathway diagrams.

*We agree with the reviewer that this was not explained sufficiently yet. We have now added explanations to all figure legends where ‘+/-p’ appear. We also revised the respective part of the Discussion section (Lines 365-370) as follows (changes in **bold**):*

“ii) The currently available canonical pathway diagrams are heterogenous in structure and, for PTM research, they may sometimes even be misleading. **One example are the phosphorylation and dephosphorylation annotations in KEGG diagrams (+/- p)**. A phosphorylating connection from kinase *A* to protein *B* just means that at least one site on *B* is a substrate of *A*, not necessarily (actually not likely) all of them, **but for a user the latter may seem implied.**”

9. Figure S6- S15: Numbers inside regulated PTM sites should be defined in legend or caption. I assume these numbers indicate counts of distinct regulated phosphosites for a given protein, but this should be explicit.

The reviewer is correct that this was not properly defined yet. They are also correct in their assumption that this indicates counts per protein and regulation category. Since the respective part of the manuscript was revised and the figures have been removed, we did not address the comment further.

10. Figure S3: Figure legend is confusing with plot.

- No unmapped gene identifiers appear on the plot. However, the legend indicates unmapped gene identifiers will be represented with a transparent gray line or fill.

- The legend also states 3,112 Uniprot accession numbers are unmapped. Are these represented in the plot?

- I suggest clarifying distinction between mapped and unmapped entries, such as by moving the 'mapped' color swatch to the left column in the legend, if correct.

We agree that the colors of the figure legend were misleading. We have updated the legend of the figure. The 3,112 Uniprot accession numbers are not represented in the plot, as indicated by the 'not shown' in the legend. Since the plot shows identifiers that were mapped to Uniprot accession numbers, the identifiers that were already in Uniprot format did not have to be mapped.

11. Figure 5 and S5: What is the explanation for the multiple MYPT1_S507 and GRLF_S589 drug potency curves? Explanation of multiple curve plotting for the same site should be mentioned explicitly in any of the main text, figure legend, or materials and methods.

We thank the reviewer for bringing this up. The reason for multiple curves for the same site is that the site was contained in multiple peptides with different sequences that were independently measured in the experiment (due to missed cleavages). We now listed the respective sequence for each curve (in the revised manuscript, the plots are part of Suppl. Figure S10).

12. Supp Fig. S6 - S12, S13-S14: Rationale for only displaying regions enclosed in dotted lines should be explicitly stated in text (either figure caption, main text or methods). Justification of the cropped zoom can help less experienced users understand how to display PTM subnetworks.

The figures in question have been removed from the manuscript as part of the revisions.

13. Figure 2: missing figure caption.

Figure 2 (now Figure 3) has been completely redone as part of the revisions. We hope that all subplots now have sufficient captions.

14. Lines 152-154 and Fig 2b: The plot in Fig 2B shows the fraction of a pathway matched between experimental and reference data. The plot clearly helps distinguish between BDNF signaling and PI3K-AKT-mTOR signaling on the basis of node matches where relevance scores show an inverse relationship. As such, analysis of 'fraction of pathway matched' such as Fig 2B may improve interpretation of g:profiler relevance scoring. Analyzing this possibility will strengthen the manuscript. Lines 152 - 154 indicate that this plot was generated from PTM Navigator. However, I could not find or may have missed this plot. Please append text to indicate how the plot was generated or link to custom code. If the plot is not already a feature of PTM Navigator, its addition is highly suggested.

We thank the reviewer for this suggestion. The plot was, in fact, not generated with PTMNavigator itself, and we did not intend to suggest that it was. In the revised version of the software, motivated also by this comment, we added a tabular display of enrichment results, including the 'fraction of

pathway/signature matched'. Therefore, the data that this plot is based on (or rather the revised version of the plot in the new Fig. 2B and 2C) can be directly shown in PTMNavigator. We considered implementing the bubble plot as well, but in the end decided against it in order not to clutter the user interface with too many visualizations.

15. Lines 74-80; 322-323: (continuation of what is mentioned in major comments) The manuscript states that PTM Navigator can be updated when better ranking procedures are developed (lines 322-323). Two alternative ranking models already exist as outlined in Krug et al. MCP, 2019. While the alternative scoring was introduced in lines 74-80, claims of superiority of g:profiler for PTM pathway analysis in lines 322-333 should be modified. If the scoring model will not be changed, at a minimum the paper should be revised to indicate g:profiler was used for its ease of implementation despite inferior performance relative to PTM-signature scoring or gene-centric-redundant scoring (Krug et al. MCP, 2019).

We apologize for the misunderstanding. As stated above, we very much value algorithms like the one by Krug et al. and did not intend to claim any superiority of g:Profiler. We have rephrased the first passage as follows (Lines 63-73 in the revised manuscript):

“While very useful for the analysis of proteome expression changes, this approach has several shortcomings in the context of PTM analysis, as has been pointed out by Krug et al. The authors introduced two algorithms that address the problems of traditional PEA: One is gene-centric redundant (GCR) single-sample gene set enrichment analysis (ssGSEA), in which genes are potentially counted multiple times during the calculation of enrichment scores, depending on how many regulated PTMs they harbor. The second is PTM-signature enrichment analysis (PTM-SEA), which uses a specifically assembled database of associations between sites of PTMs and perturbations, pathways, or kinase activities (PTM signatures database; PTMsigDB) and thus can calculate enrichments on PTM level. This constitutes a valuable complementary approach to conventional PEA and both methods have shown superior performance compared to non-redundant gene-centric PEA.”

The second passage was not changed, but we hope that in the context of the revised manuscript it now will not be interpreted anymore as a claim of superiority for g:Profiler.

Reviewer #3 (Remarks to the Author):

We thank the reviewer for reading and co-reviewing our manuscript.

Reviewer #4 (Remarks to the Author):

Summary

The authors describe in this manuscript a tool for visualizing phosphoproteomics data within

curated pathway diagrams from KEGG and WikiPathways. The tool can visualize qualitative or quantitative phosphosite changes, and integrates relevant information from PhosphoSitePlus and other sources to facilitate the interpretation of these changes. The authors demonstrate the utility of this tool through several case studies: 1) it can facilitate manually discarding enriched pathways that do not seem biologically plausible; 2) it helps reveal the flow of signal through pathways and differences in signal flow between perturbations; 3) it expedites the process of proposing new links in pathways based on associations in omics data. While this is a useful resource for the signaling and proteomics communities, the tool and the accompanying manuscript are somewhat limited in their scope and may be more suitable for a more specialized audience.

We thank the reviewer for their positive feedback and general assessment of our study. Before we begin our point by point response, we would like to outline the most notable additions we made to our software: i) Pathway Editor: Users can now create new or modify existing pathway diagrams by adding and removing nodes and edges, as well as rearranging them on the canvas. ii) Enrichment Server: We implemented a service application that hosts a number of pathway and kinase enrichment analysis algorithms for phosphoproteomics data such as ssGSEA, KSEA, ROKAI, PHONEMeS, Kinase library and KEA3. Using this service, we now provide new options for sorting pathways by putative relevance such as Gene-Centric Redundant ssGSEA Scores. iii) Data-driven putative kinase signaling pathways: Based on the output of kinase enrichment algorithms, we generate a data-driven putative kinase signaling pathway using the PHONEMeS tool. This tool uses a prior knowledge network of protein-protein interaction and kinase-substrate relationships to reconstruct a signaling network, which can subsequently be viewed and edited in PTMNavigator. iv) Context Menu for customization of visualizations: We added a context-menu ('right-click menu'), which incorporates several new optional visualization features such as highlighting enriched kinases, labeling PTM Nodes with their amino acid residue and position and displaying kinase-substrate relationships from PhosphoSitePlus as additional edges.

Major concerns

While I believe it is useful to have a way to draw pathways with quantitative information from phosphoproteomics studies, the major concern I have is that, as of yet, the tool does not facilitate the derivation of novel information. Furthermore, the advance made either in terms of method/tool development or biological discoveries remains limited. For the tool to comprehensively facilitate pathway-level analysis of phosphoproteomics data, it would benefit from incorporating additional information and common analyses. For the benefit of the authors I include below some example analysis that would improve the usefulness of a such a tool:

We agree that the first version of PTMNavigator was primarily a visualization tool with some additional benefits like the PhosphoSitePlus integration. With the newly implemented features, most notably (i) the creation of custom pathway diagrams, (ii) the automatically performed enrichment analyses, and (iii) the automatic creation of a data-driven pathway diagram using PHONEMeS, we argue that our tool now enables in-depth analysis of PTM data.

1) Performing differential expression analysis. Currently the user must do this themselves, which slows down the overall process.

We have considered this option, but, in the end, consciously abstained from adding it. There are many different ways to perform differential expression analysis, each with their own set of parameters and options. Most users will already have tools and software that they are familiar with that can perform such tasks (e.g., Perseus, limma, or SciPy). Our vision for PTMNavigator is that it leaves this freedom (and therefore also the responsibility) to the user.

2) Incorporating kinase-substrate predictions from tools such as NetworKIN and the Kinase Library (10.1093/nar/gkm902, 10.1038/s41586-022-05575-3). Since the majority of differentially regulated phosphosites in a given phosphoproteomics experiment will have no annotated upstream kinase, these predictions can greatly facilitate the interpretation of results. This would be especially true when results are visualized within a pathway diagram, where predicted kinases can be viewed within their pathway context.

We agree with the reviewer's comment and now integrated the data from the kinase library in a 'motif-enrichment' manner but decided against adding it in a 'kinase-substrate-prediction' manner. We found that the top scoring kinases provided for individual p-sites as provided in the PhosphositePlus interface often did not match our expectations. The expected kinases typically appeared in the first page but not among the top scoring kinases. Providing such information in PTMNavigator could give users the misleading idea that these predictions are reliable. Nevertheless, users can still access this information by following the link to PhosphositePlus in the tooltip. Additional work is needed to find good ways of incorporating such kinase-substrate predictions in pathway diagrams but this is outside the scope of the current manuscript. We did not integrate NetworKIN since, in our experience, the predictions from the Kinase Library were more informative.

Our newly added context menu (see above) provides the option to visualize high-confidence kinase-substrate relationships from PhosphoSitePlus. Note that visualizing all top candidate predictions within the pathway diagram would make the visualization cluttered and possibly misleading for the reasons mentioned above.

3) Incorporating estimated changes in kinase activity using tools such as KSEA, KEA3, etc. (10.1093/bioinformatics/btx415, 10.1093/nar/gkab359). In the current implementation of the tool, users can only infer pathway signal flow by tracking changes in individual phosphosites. Given kinases are the major conduits for signal flow, estimates of their regulation would be useful.

This is indeed a very useful suggestion and we have integrated it. For every uploaded dataset, kinase activity enrichment is now automatically performed using KSEA, KEA3, RoKAI in combination with KSEA (<https://www.nature.com/articles/s41467-021-21211-6>), and KSTAR (<https://www.nature.com/articles/s41467-022-32017-5>). In addition, the motif enrichment can also be seen as a form of kinase activity estimation. The results of the RoKAI+KSEA enrichment, which we consider the most robust estimation of kinase activity, can now also be visualized within the pathway view of PTMNavigator.

The pathway diagrams contain directed edges from kinases to other proteins, indicating that the kinase phosphorylates a subset of the protein's phosphosites. The authors have provided access to this information through a tooltip, which summarizes PhosphositePlus annotation including upstream kinases. However, it would be more informative if specific kinase-phosphosite relationships were incorporated as edges within the diagram itself. This would allow the user to immediately track signal flow from kinases to their substrates; currently they must manually click on each substrate phosphosite to do this.

We thank the reviewer for this suggestion and have implemented this functionality in the new version of our tool.

In the introduction the authors highlight several limitations of pathway enrichment analysis, such as that it does not distinguish between randomly distributed changes vs discernible signal flow, and that it considers all genes as equally important to pathway activity. While the tool helps tackle these problems, it does not overcome them in a systematic manner; the burden is still on the researcher to manually inspect and judge enriched pathways.

This is indeed correct. Automated assessment of such issues is, unfortunately, a complex task due to the heterogeneity and limited coverage of pathways in current databases. Still, we are confident that PTMNavigator greatly reduces the effort needed to perform such analyses.

Minor points

It would be helpful if phosphosites were automatically labeled with their residue and position (e.g. Y187 on MAPK1). Currently the user has to manually click on phosphosites to see this information.

We thank the reviewer for this suggestion. As mentioned above, we have implemented this functionality in the new version of our tool. Note that we made this feature optional, to avoid overloading the user interface at the start.

The authors suggest that PTMNavigator can be used to uncover unknown relationships between PTMs and pathway components, and hence extend existing pathways (e.g. lines 24-26 in the abstract). The unknown relationships that they propose (e.g. an unknown interaction between MAP2K1/2 and GRLF1) are based on correlations in phosphoproteomics data. Without additional evidence for these relationships (for example in vitro kinase assays showing MAP2K1/2 can phosphorylate GRLF1) I would be hesitant to extend a pathway diagram in such a way. I recommend that the authors weaken the language used, for example by saying that PTMNavigator “can aid in proposing putative new links in pathways”.

We thank the reviewer for this assessment. We integrated the statement suggested by the reviewer into the Abstract as follows (Lines 30-32):

“Third, PTMNavigator aided in proposing extensions to an existing pathway by suggesting putative new links between both PTMs and pathway components.”

Point to point response to reviewer comments

Reviewer #1 (Remarks to the Author):

The authors have addressed many of the points raised during the first round of revision, this has strengthened PTMNavigator as a tool for the community. After engaging with the website I have several remaining requests, comments and concerns.

We thank the reviewer for the positive feedback and for acknowledging that the tool and manuscript have been strengthened.

1. When loading my own phosho dataset, I followed the format suggestion provided in 'examples'>'fold_change_data'>'psite_level_data.csv'. I was unable to see any enrichment scores for pathways, no PHONEMeS custom pathway generation, nor any scoring with other analysis tools in my uploaded data. If this is a result of formatting issues, or insufficient input data, this is unclear to me as a user. If the analysis was not yet complete, the status and progress of the task was unclear. Given my struggle to use the tool successfully, I suggest more clarity is provided to the user to abide by necessary formatting. More troubleshooting direction would be helpful beyond the provided text "Did you just upload this dataset? If it still is not here after a few minutes, something probably went wrong. Please contact a site administrator."

We apologize for this issue. In fact, the functionality of automatic enrichment analyses was, until now, only enabled for peptide- and protein-level data, but not for p-site data, and we did not point this out anywhere. However, we have now implemented this feature for all enrichment algorithms provided by our service, with the exception of K-STAR, which requires peptide sequences as input (we added this information to the frontend, too). Therefore, we suggest the reviewer try the upload of their dataset again.

Furthermore, we'd like to point out that in the data upload interface, there are indeed several error messages if the data does not comply with the expected format - probably the reviewer did not encounter them, since their data were already correctly formatted.

2. I could not successfully load any datasets from the ProteomicsDB_Data list. This was true across different browsers and for different projects listed.

We also apologize for this issue. The cause of this were unexpected problems with the hardware and software of ProteomicsDB in July and August 2024, which brought down the entire site for several weeks, and also after the site was back online, some of the platform's functionality remained broken. Most features, including this, are repaired now, so we again suggest the reviewer try this feature out once more.

3. It is helpful to see analysis tools like PTM SEA integrated into the browser, however there is no associated metadata available for the user to see details on how the enrichment analysis was performed. For example, in PTM sea there are several parameters the user can adjust including normalization method, overlap minimums, weighting, etc. PTMNavigator should provide information equivalent to a parameters output file for the enrichments that are performed on the datasets. This is helpful for FAIR data purposes. Furthermore, can the list of protein/geneIDs that are actually being used in the enrichment be added to the data tables? This way users can see directly which protein nodes are contributing to the enrichment score. They can then use these

nodes as filtering criteria for their pathway searches using the provided 'Filter pathways by Protein List' feature.

Following these suggestions, we added:

1. A button on top of each enrichment table that displays a dialog with the parameter values used, or other details about how the analysis was carried out (Revision Figure 1).
2. A column to the ssGSEA tables (PTM-SEA, GC, GCR) that shows the overlap between the data and each gene/PTM signature (Revision Figure 2).

Revision Figure 1: Parameter files and implementation details can now be displayed in PTMNavigator.

Signature ID	Overlap (MyExperiment)	Percent Overlap ...	adj p-val...	Score (MyExperiment) ↓
WP_FOCAL_ADHESION	PAK2 BCAR1 MYL12B SHC2 PRKCA T...	29.1%	0.02	-5.98
KEGG_FOCAL_ADHESION	PAK2 BCAR1 MYL12B SHC2 PRKCA V...	28.1%	0.02	-5.94
WP_PHYSICO_CHEMICAL_FEATURES...	PFN1 MAP2K1 ROCK2 AXIN1 APC PL...	37.9%	0.02	-4.91
WP_THYROID_HORMONES_PRODUCT...	TBC1D4 MAP4K5 PRKACA RPS6KA6 ...	29%	0.02	-4.77
WP_NANOPARTICLE_MEDIATED_ACTI...	MAP2K1 MAP2K2 ITGB1 PTK2 RAF1 ...	46.4%	0.02	-4.73

Revision Figure 2: A pathway enrichment table in PTMNavigator with the newly added 'Overlap' column.

4. The Expand function in the tool would be more useful if there were additional levels of selectivity beyond 'expand all'. For example, can the user only expand regulated sites, and/or sites for only the selected node(s)? Subsetting the data before expansion is a useful feature, especially for nodes with many PTM sites.

This is helpful feedback and we addressed it in the present version of PTMNavigator. When right-clicking on the canvas, it is now possible to expand only the regulated or only the selected sites, in addition to all sites in the diagram. Moreover, when right-clicking on a protein node in the graph, users now have the option to expand only the PTM nodes associated with this node (all or only the regulated ones).

5. It is not clear how multiple studies are integrated into the pathway maps or used to generate the pathway scores. Are multiple datasets or experiments considered when generating the pathway enrichment scores? If yes, how is this done? How can users readily see which sites are being mapped from which experiment? There should be more features added to understand when the user is viewing combined data or data from a single experiment, and how data integration is occurring.

This was in fact a shortcoming of PTMNavigator so far. We extended our software in several ways in order to improve this:

- 1. We added a dataset selection menu on top of the enrichment analysis tables (Revision Figure 3). For each dataset, the enrichment analyses are performed and displayed separately.*
- 2. We added another dataset selection menu above the pathway selection menu (Revision Figure 4). This determines which dataset is used for sorting the pathway list by enrichment score. If a dataset consists of multiple experiments, the scores are summed across experiments for sorting.*
- 3. For the highlighting of kinase activities in the diagram, we added a new component where users can choose the dataset and the activity inference method (KSEA, RoKAI, KSTAR, Motif Enrichment, KEA3), and adjust thresholds for the visualization (score & significance) (Revision Figure 5).*
- 4. In the graph, the dataset and experiment of each PTM node are visible in the node's tooltip. We moved this information up in the list of details, so it is easier to see. The new table view (see next comment) also includes the dataset and experiment for each PTM node.*

Revision Figure 3: Dataset selection on top of enrichment analyses tables.

Sort Pathways by Dataset Enrichment:
 BekkerJensen2020_MEKi_Cobi... ▼

Pathways (n=1121): ▼

Revision Figure 4: Dataset selection on top of canonical pathway list.

Highlight Kinase Activities:

Show Kinase Activities for Dataset:
 BekkerJensen2020_MEKi_Cobi... ▼

Kinase Activity Inference Method
 RoKAI+KSEA ▼

Set thresholds:

$|Score| \geq$ 0.5

$-\log_{10}(pval) \geq$ 0.2

Revision Figure 5: The new component that allows to control the parameters of the kinase activity highlighting in the graph.

6. At present it is difficult to see all of the phosphosites associated with a protein node. If the user clicks on the node, no PTM data is listed in the 'currently selected tab.' Is it possible to add a table view to the current selection where all of the associated phosphosites for a selected protein node are listed? This would also be a place to directly see, in table form, the PTM regulation occurring from different experiments that are loaded into the pathway.

This was another useful feedback. We have added two new tables to the user interface which dynamically show the sites and proteins that are selected in the graph, including all details that are in the tooltips. These tables are located below the graph, at the same position as the enrichment

tables – the user can toggle between the three table views (Revision Figure 6). All tables can be downloaded as CSV files.

Regulation	Dataset	Modified Sequence	Gene Name(s)	Uniprot	Site(s)	Upstream Kinase(s)
↑	BekkerJensen2020_MEKI_Cobimetinib	SNS(ph)LRRDS(ph)PPPPAR	PAK4	O96013	S99 & S104	PRKD1, PRKD2
↑	BekkerJensen2020_MEKI_Cobimetinib	RRS(ph)LEPAENVHGAGGGAFPASQTPSK	SRC	P12931	S17	
↓	BekkerJensen2020_MEKI_Cobimetinib	TSSVSNPQDSVGS(ph)PCSR	PXN	P49023	S106	
↑	BekkerJensen2020_MEKI_Cobimetinib	TGSSS(ph)PPGGPPKPGSQLDSMLGSLQSDLNK	PXN	P49023	S322	
—	BekkerJensen2020_RAFI_Sorafenib	RRS(ph)LEPAENVHGAGGGAFPASQTPSK	SRC	P12931	S17	
↑	BekkerJensen2020_RAFI_Sorafenib	TGSSS(ph)PPGGPPKPGSQLDSMLGSLQSDLNK	PXN	P49023	S322	
↓	BekkerJensen2020_RAFI_Sorafenib	QKS(ph)AEPS(ph)PTVMSTSLGSLNSELDR	PXN	P49023	S126 & S130	MAPK3, MAPK1

Revision Figure 6: The newly added 'Selected peptides' table.

7. The figures displayed in Figure 3B/C/D were generated with other features of ProteomicsDB, correct? If so, the authors should indicate in the manuscript which ProteomicsDB analysis features were used. If the data analysis is outside the scope of ProteomicsDB, then the manuscript should state that the analysis was done independent of the webtool.

The figures were not generated with ProteomicsDB, but with Python (Plotly package), using the data provided in the supplementary tables. This is pointed out in the figure legend now. We considered the option of adding bubble plots to PTMNavigator, but ultimately decided against it for the purpose of not overloading the user interface with too many visualizations.

Reviewer #2 (Remarks to the Author):

The revised manuscript incorporates major changes asked by reviewers. We appreciate the work invested to incorporate kinase activity inference and pathway scoring algorithms, as well as improve flexibility of the visualizations. We are satisfied with the revised version and suggest proceeding with publication.

We thank the reviewer for the positive feedback and for the endorsement for publication.

That said, we noticed a few minor technical issues with usability of the new features, listed below:
1. Contrary to the provided instructions, CTRL click seems to deselect nodes from the view. If this is supposed to be the functionality, please make the instructions clearer.

The reviewer here indeed discovered an unexpected behavior that we did not notice before. CTRL+Click does indeed add nodes to the current selection, but it also can remove a node from the current selection. This is the same behavior as in other software, e.g. when selecting files in the Windows Explorer. However, in the initial state of a PTMNavigator graph, **all** nodes are considered "selected", so if CTRL is pressed at the first Click already, it would lead to a deselection instead of

a selection of the clicked node (and we assume this is what the reviewer attempted to do). We agree that this is counterintuitive, and changed this behavior: The first click now always leads to a selection (and deselection of the rest of the graph), regardless of whether CTRL is pressed or not. On subsequent clicks, pressing CTRL adds to/removes from the current selection. We hope that the behavior is more intuitive now.

2. Kinase relationships shown by red arrows only appear if the user selects 'Expand All' in the right-click menu in addition to 'Show kinase and substrate relationships'. Please add this to the instructions on the site or fix this default.

This is helpful feedback regarding the user-friendliness of our tool. In the updated version, when clicking 'Show kinase and substrate relationships' as well as 'Show PTM labels', all affected nodes are automatically expanded to show the information straight away.

3. Is it possible to avoid having the figure disappear when transitioning between viewing and editing modes? It is somewhat inconvenient to have it disappear when switching modes. We recommend these three issues are fixed prior to publication.

This was undesired behavior and we fixed it. When switching from viewing to editing mode, the currently loaded datasets are now preserved internally (but not displayed anymore because the edit mode is dataset-independent). When switching back to viewing mode, the preserved data is projected again onto whichever pathway diagram is visible at that time.

Reviewer #3 (Remarks to the Author):

Reviewer #3 (Remarks on code availability):

I reviewed the code in the first version of the manuscript. I checked the output of the new features worked as expected.

There is a readme in the features code.

We thank the reviewer for evaluating the manuscript and functionality of our updated software.

Reviewer #4 (Remarks to the Author):

I appreciate the work of the authors in revising the manuscript. They addressed the majority of my suggestions and have substantially enhanced the functionality and usability of the tool, making it a better resource for the research community. Some notable improvements include the addition of edges of kinase-substrate interactions, phosphosite labels, and the premise of automatically performing pathway enrichment and ranking pathways using ssGSEA.

We thank the reviewer for taking the time in evaluating the updated versions of manuscript and software, and for the favorable feedback.

While testing the tool I did encounter some limitations with the new features. First, the ssGSEA

enrichment seems to be buggy or errorprone. For example, I loaded one of the supplied datasets (Session ID: 0123456789ABCDEF0123456789ABCDEF, uploaded datasets: BekkerJensen2020_AKTi_AKT-1-2, Experiment Filter: BekkerJensen2020_AKTi_AKT-1-2) and selected the most highly ranked pathway (3q29 copy number variation syndrome (WP4906) [Score=4.38]). However, on inspection of the pathway I could not find any differentially regulated phosphosites, which seems in direct contradiction with it being the most highly ranked pathway.

ssGSEA enrichment only takes fold changes into account, not significance of regulation. It is possible to use only significantly regulated data points as input for the enrichment, but in the case of this dataset, this did not return any significant overlap with MSigDB signatures. Based on the fold changes in the AKT-1-2 dataset, the pathway with the highest normalized enrichment score and lowest p-value in gene-centric redundant GSEA is indeed WP4906 – ‘3q29 copy number variation syndrome’ (see also the file 4_BekkerJensen_GCR.xlsx of the additional supplementary tables, available here: <https://doi.org/10.5281/zenodo.12720528>). The adjusted p-value, however, is only 0.13, indicating that there are no really significant pathways enriched in the dataset. PTM-SEA analysis (also ssGSEA-based) suggests a strong and significant negative enrichment of the CSNK2A2 signature, a hypothesis that is also supported by the KSEA and RoKAI kinase activity enrichment results.

Second, while it is useful to display kinase activity estimates on the pathway diagram, these were visually not very salient; furthermore, it was unclear to me what threshold had been used to select active kinases. This could be displayed either in the tool, or the user could be allowed to select the threshold themselves. Given that multiple kinase enrichment tools have now been provided in PTMNavigator, it would also make sense to allow the user to select the specific source of kinase enrichment.

We agree that in hindsight, hiding the details on the kinase activity highlighting from the user was not a good choice (we did it for the sake of simplicity). We now implemented an additional component into our tool to address this request (Revision Figure 7). With this component, users can choose which kinase activity inference algorithm should be used to generate the lists of perturbed kinases, and they can also set thresholds on absolute score and significance. If multiple datasets are loaded, they can select which one should be used for the enrichment.

Highlight Kinase Activities:

Show Kinase Activities for Dataset:
BekkerJensen2020_MEKi_Cobi... ▼

Kinase Activity Inference Method
RoKAI+KSEA ▼

Set thresholds:

$|Score| \geq$  0.5

$-\log_{10}(pval) \geq$  0.2

Revision Figure 7: The new component that allows to control the parameters of the kinase activity highlighting in the graph.

Importantly, the tool still does not address a critical issue in pathway enrichment analysis: how to systematically prioritize pathways with discernable signal flow, rather than those with randomly distributed changes. The tool only provides means for a user to attempt this by manual inspection. Overall, while the authors have substantially improved their tool and the accompanying manuscript, I believe this represents a modest methodological and biological advance.

This is a valid observation. Tackling the systematic prioritization of relevant versus random pathway hits turned out to be beyond the scope of this project; more sophisticated pathway enrichment algorithms will be required for this. However, we believe that the present manuscript and software already represent a step in this direction that will be welcomed by the community.

The authors thank the reviewers for reading and evaluating our revised software and manuscript. Based on their comments, we made further extensions to PTMNavigator, which we believe enhance the user-friendliness and clarity of the interface, and we hope that the tool and manuscript are now fit for publication. Please find our point by point responses to the reviewers below.

With best wishes,

Bernhard Kuster & Matthew The

Point to point response to reviewer comments

Reviewer #1 (Remarks to the Author):

I am very happy to read that the authors considered all the suggestions and updated the website accordingly. These edits, once successfully implemented, will help make PTM navigator a more clear useful tool. I am eager to see this webtool operate to its full potential as this is a tool I would recommend to my lab and collaborators.

We thank the reviewer for the positive feedback and for acknowledging that the tool and manuscript have been strengthened.

That being said, I was unable to experience some of the improvements mentioned in the rebuttal. For example, no where could I find any enrichment for any dataset, not my own data or the preloaded dataset files. Previously I could see scoring in the pathways and in the Enrichment Analysis with the provided datasets, I just couldn't get it to work with my own data. Now I see no enrichment anywhere with any dataset. The only thing that happened during my interaction was a list of all possible pathways was loaded, with no scoring. I could not find the Figure 4 revision with threshold scoring anywhere on the site. If perhaps this is my own user error, I spend 30 min attempting to figure it out with no success, so the intuitive user experience may need to be addressed. Without any scoring, the entire utility of the tool is decreases as now I have 1000+ unranked pathways to sort from to visualise my data. I therefore would not recommend or use the tool in its present state.

We'd like to thank the reviewer for taking the time to evaluate our tool so thoroughly and put together the screenshots (sent in a separate email). This helped us to isolate the cause of one misunderstanding:

With "preloaded datasets" we referred to the datasets accessible under "User data" with the Session ID 0123456789ABCDEF0123456789ABCDEF (the datasets that are discussed in the paper). The reviewer was instead referring to the "ProteomicsDB Data" datasets for which enrichment results were indeed not available. This was an oversight on our part and enrichments for these datasets have now been added (since Nov 4th).

Unfortunately, we are unable to reproduce the custom dataset issue. It should indeed work for both fold change (single-dose) data as well as dose-dependent data.

Nevertheless, we made several updates that improve user friendliness and may resolve the reviewer's issue:

1. we added status indicators for each enrichment analysis (see attached photo). Green: enrichment successfully completed, yellow: still running, red: algorithm failed, grey: "not applicable" (e.g. Kinase Activity Inference for non-phospho datasets).
2. the interface is automatically refreshed to reflect the enrichment analysis results as soon as they are completed. This includes the status indicators, the sorted pathway dropdown list and the enrichment result tables.
3. the enrichment algorithms are now run in parallel. This decreases the runtime from ~10 minutes to ~3-4 minutes on the 10k peptide example dataset. Only the KSTAR enrichments are still very slow, we are investigating this and have decided to disable them for the time being, so as not to stall the other analyses.
4. we added information boxes with the typical enrichment runtime and a warning that pathways are not sorted until the PEA-GCR enrichment result has finished.

We have asked our colleagues to test these functionalities and they were able to successfully run their analyses.

Revision_Figure 1: The newly added component that shows the status of each enrichment analysis of the currently loaded dataset.

An additional suggestion, I find myself frequently lost in scrolling to try and find the pathway that has floated away in the zoom space. Is there some way to recover the original zoom and x y z placement of the image with a right click?

We thank the reviewer for this suggestion. We haven't implemented this yet, but we put it on a ToDo-list of features that we want to add in the near future.